# UNIFIED PRINCIPLES FOR MULTI-SOURCE TRANSFER LEARNING UNDER LABEL SHIFTS

## ABSTRACT

We study the label shift problem in multi-source transfer learning and derive new generic principles. Our proposed framework unifies the principles of conditional feature alignment, label distribution ratio estimation and domain relation weights estimation. Based on inspired practical principles, we provide unified practical framework for three multi-source label shift transfer scenarios: learning with limited target data, unsupervised domain adaptation and label partial unsupervised domain adaptation. We evaluate the proposed method on these scenarios by extensive experiments and show that our proposed algorithm can significantly outperform the baselines.

## 1 INTRODUCTION

Transfer learning (Pan & Yang, 2009) is based on the motivation that learning a new task is easier after having learned several similar tasks. By learning the inductive bias from a set of related source domains $(\mathcal{S}_1, \ldots, \mathcal{S}_T)$ and then leveraging the shared knowledge upon learning the target domain $\mathcal{T}$, the prediction performance can be significantly improved. Based on this, transfer learning arises in deep learning applications such as computer vision (Zhang et al., 2019; Tan et al., 2018; Hoffman et al., 2018b), natural language processing (Ruder et al., 2019; Houlsby et al., 2019) and biomedical engineering (Raghu et al., 2019; Lundervold & Lundervold, 2019; Zhang & An, 2017).

To ensure a reliable transfer, it is critical to understand the theoretical assumptions between the domains. One implicit assumption in most transfer learning algorithms is that the label proportions remain unchanged across different domains (Du Plessis & Sugiyama, 2014) (i.e., $\mathcal{S}(y) = \mathcal{T}(y)$). However, in many real-world applications, the label distributions can vary markedly (i.e. label shift) (Wen et al., 2014; Lipton et al., 2018; Li et al., 2019b), in which existing approaches cannot guarantee a small target generalization error, which is recently proved by Combes et al. (2020).

Moreover, transfer learning becomes more challenging when transferring knowledge from *multiple sources* to build a model for the target domain, as this requires an effective selection and leveraging the most useful source domains when label shift occurs. This is not only theoretically interesting but also commonly encountered in real-world applications. For example, in medical diagnostics, the disease distribution changes over countries (Liu et al., 2004; Geiss et al., 2014). Considering the task of diagnosing a disease in a country without sufficient data, how can we leverage the information from different countries with abundant data to help the diagnosing? Obviously, naïvely combining all the sources and applying one-to-one single source transfer learning algorithm can lead to undesired results, as it can include low quality or even untrusted data from certain sources, which can severely influence the performance.

In this paper, we study the label shift problem in multi-source transfer learning where $\mathcal{S}_t(y) \neq \mathcal{T}(y)$. We propose *unified principles* that are applicable for three common transfer scenarios: unsupervised Domain Adaptation (DA) (Ben-David et al., 2010), limited target labels (Mansour et al., 2020) and partial unsupervised DA with $\text{supp}(\mathcal{T}(y)) \subseteq \text{supp}(\mathcal{S}_t(y))$ (Cao et al., 2018), where prior works generally treated them as separate scenario. It should be noted that this work deals with target shift *without* assuming that semantic conditional distributions are identical (i.e., $\mathcal{S}_t(x|y) \neq \mathcal{T}(x|y)$), which is more realistic for real-world problems. Our contributions in this paper are two-folds:

(**I**) We propose to use Wasserstein distance (Arjovsky et al., 2017) to develop a new target generalization risk upper bound (Theorem 1), which reveals the importance of label distribution ratio

estimation and provides a principled guideline to learn the domain relation coefficients. Moreover, we provide a theoretical analysis in the context of representation learning (Theorem 2), which guides to learn a feature function that minimizes the conditional Wasserstein distance as well as controls the weighted source risk. We further reveal the relations in the aforementioned three scenarios lie in the *different assumptions* for estimating label distribution ratio.

(**II**) *Inspired* by the theoretical results, we propose *Wasserstein Aggregation Domain Network* (WADN) for handling label-shift in multi-source transfer learning. We evaluate our algorithm on three benchmark datasets, and the results show that our algorithm can significantly outperform state-of-the-art principled approaches.

## 2    RELATED WORK

**Multi-Source Transfer Learning Theories**   have been investigated in the previous literature with different principles to aggregate source domains. In the popular unsupervised DA, (Zhao et al., 2018; Peng et al., 2019; Wen et al., 2020; Li et al., 2018b) adopted $\mathcal{H}$-divergence (Ben-David et al., 2007), discrepancy (Mansour et al., 2009) and Wasserstein distance (Arjovsky et al., 2017) of marginal distribution $d(\mathcal{S}_t(x), \mathcal{T}(x))$ to estimate domain relations and dynamically leveraged different domains. These algorithms generally consists source risk, domain discrepancy and an un-observable term $\eta$, the optimal risk on all the domains, which are ignored in these approaches. However, as Combes et al. (2020) pointed out, ignoring the influence of $\eta$ will be problematic when label distributions between source and target domains are significantly different. Therefore it is necessary to take $\eta$ into consideration by using a small amount of labelled data is available for the target domain (Wen et al., 2020). Following this line, very recent works (Konstantinov & Lampert, 2019; Wang et al., 2019a; Mansour et al., 2020) started to consider measure the divergence between two domains given label information for the target domain by using $\mathcal{Y}$-discrepancy (Mohri & Medina, 2012). However, we empirically showed these methods are still unable to handle label shift.

**Label-Shift**   Label-Shift (Zhang et al., 2013; Gong et al., 2016) is a common phenomena in the transfer learning with $\mathcal{S}(y) \neq \mathcal{T}(y)$ and generally ignored by the previous multi-source transfer learning practice. Several theoretical principled approaches have been proposed such as (Azizzadenesheli et al., 2019; Garg et al., 2020). In addition, (Combes et al., 2020; Wu et al., 2019) analyzed the generalized label shift problem in the one-to-one single unsupervised DA problem but did not provide guidelines of *levering different sources* to ensure a reliable transfer, which is more challenging. (Redko et al., 2019) proposed optimal transport strategy for the multiple unsupervised DA under label shift by assuming identical semantic conditional distribution. However they did not consider representation learning in conjunction with their framework and did not design neural network based approaches. Different from these, we analyzed our problem in the context of representation learning and propose an efficient and principled strategies. *Moreover, our theoretical results highlights the importance of label shift problem in a variety of multi-source transfer problem.* While the aforementioned work generally focus on the unsupervised DA problem, without considering unified rules for different scenarios (e.g. partial multi-source DA).

## 3    THEORETICAL INSIGHTS: TRANSFER RISK UPPER BOUND

We assume a scoring hypothesis defined on the input space $\mathcal{X}$ and output space $\mathcal{Y}$ with $h : \mathcal{X} \times \mathcal{Y} \to \mathbb{R}$, is $K$-Lipschitz w.r.t. the feature $x$ (given the same label), i.e. for $\forall y$, $\|h(x_1, y) - h(x_2, y)\|_2 \leq K\|x_1 - x_2\|_2$, and the loss function $\ell : \mathbb{R} \times \mathbb{R} \to \mathbb{R}_+$ is positive, $L$-Lipschitz and upper bounded by $L_{\max}$. We denote the expected risk w.r.t distribution $\mathcal{D}$: $R_{\mathcal{D}}(h) = \mathbb{E}_{(x,y)\sim\mathcal{D}}\ell(h(x,y))$ and its empirical counterpart (w.r.t. $\hat{\mathcal{D}}$) $\hat{R}_{\mathcal{D}}(h) = \sum_{(x,y)\in\hat{\mathcal{D}}} \ell(h(x,y))$. We adopted Wasserstein-1 distance (Arjovsky et al., 2017) as a metric to measure the similarity of the domains. Compared with other divergences, Wasserstein distance has been theoretically proved tighter than TV distance (Gong et al., 2016) or Jensen-Shnannon divergence (Combes et al., 2020).

Based on previous work, the label shift is generally handled by label-distribution ratio weighted loss: $R_{\mathcal{S}}^\alpha(h) = \mathbb{E}_{(x,y)\sim\mathcal{S}}\alpha(y)\ell(h(x,y))$ with $\alpha(y) = \mathcal{T}(y)/\mathcal{S}(y)$. We also denote $\hat{\alpha}_t$ as its empirical counterpart, estimated from samples. Besides, to measure the task relations, we define a simplex $\boldsymbol{\lambda}$ with $\boldsymbol{\lambda}[t] \geq 0, \sum_{t=1}^T \boldsymbol{\lambda}[t] = 1$ as the *task relation coefficient* vector by assigning high weight to

the similar task. Then we first present Theorem 1, which proposed theoretical *insights* about how to combine source domains through properly estimating $\boldsymbol{\lambda}$.

**Theorem 1.** *Let $\{\hat{\mathcal{S}}_t = \{(x_i, y_i)\}_{i=1}^{N_{\mathcal{S}_t}}\}_{t=1}^{T}$ and $\hat{\mathcal{T}} = \{(x_i, y_i)\}_{i=1}^{N_{\mathcal{T}}}$, respectively be $T$ source and target i.i.d. samples. For $\forall h \in \mathcal{H}$ with $\mathcal{H}$ the hypothesis family and $\forall \boldsymbol{\lambda}$, with high probability $\geq 1 - 4\delta$, the target risk can be upper bounded by:*

$$R_{\mathcal{T}}(h) \leq \sum_t \boldsymbol{\lambda}[t] \hat{R}_{\mathcal{S}_t}^{\hat{\alpha}_t}(h) + LK \sum_t \boldsymbol{\lambda}[t] \mathbb{E}_{y \sim \hat{\mathcal{T}}(y)} W_1(\hat{\mathcal{T}}(x|Y=y) \| \hat{\mathcal{S}}_t(x|Y=y)) + L_{\max} d_\infty^{\sup} \sqrt{\sum_{t=1}^{T} \frac{\boldsymbol{\lambda}[t]^2}{\beta_t}} \sqrt{\frac{\log(1/\delta)}{2N}}$$

$$+ L_{\max} \sup_t \|\alpha_t - \hat{\alpha}_t\|_2 + Comp(N_{\mathcal{S}_1}, \dots, N_{\mathcal{S}_T}, N_{\mathcal{T}}, \delta),$$

*where $N = \sum_{t=1}^{T} N_{\mathcal{S}_t}$ and $\beta_t = N_{\mathcal{S}_t}/N$ and $d_\infty^{\sup} = \max_{t \in [1,T], y \in [1,\mathcal{Y}]} \alpha_t(y)$ the maximum true label distribution ratio value. $W_1(\cdot\|\cdot)$ is the Wasserstein-1 distance with $L_2$-distance as cost function. $Comp(N_{\mathcal{S}_1}, \dots, N_{\mathcal{S}_T}, N_{\mathcal{T}}, \delta)$ is a function that decreases with larger $N_{\mathcal{S}_1}, \dots, N_{\mathcal{S}_T}$, given a fixed $\delta$ and hypothesis family $\mathcal{H}$. (See Appendix E for details)*

**Remarks** (1) In the first two terms, the relation coefficient $\boldsymbol{\lambda}$ is controlled by $\alpha_t$-weighted loss $\hat{R}_{\mathcal{S}_t}^{\hat{\alpha}_t}(h)$ and conditional Wasserstein distance $\mathbb{E}_{y \sim \hat{\mathcal{T}}(y)} W_1(\hat{\mathcal{T}}(x|Y=y) \| \hat{\mathcal{S}}_t(x|Y=y))$. To minimize the upper bound, we need to assign a higher $\boldsymbol{\lambda}[t]$ to the source $t$ with a *smaller weighted prediction loss* and *a smaller weighted semantic conditional Wasserstein distance*. Intuitively, we tend to leverage the source task which is semantic similar to the target and easier to learn. (2) If each source have equal observations with $\beta_t = 1$, then the third term will become $\|\boldsymbol{\lambda}\|_2$, a $L_2$ norm regularization, which can be viewed as an encouragement of uniformly leveraging all the sources. Combing these three terms, we need to consider the trade-off between assigning a higher $\boldsymbol{\lambda}[t]$ to the source $t$ that has a smaller weighted prediction loss and conditional Wasserstein distance, and assigning balanced $\boldsymbol{\lambda}[t]$ for avoiding concentrating on only one source. (3) $\|\hat{\alpha}_t - \alpha_t\|_2$ indicates the gap between ground-truth and empirical label ratio. Therefore if we can estimate a good $\hat{\alpha}_t$, these terms can be small. In the practice, If target labels are available, $\hat{\alpha}_t$ can be computed from the observed data and $\hat{\alpha}_t \rightarrow \alpha_t$. If target labels are absent (unsupervised DA), we need to design methods and to properly estimate $\hat{\alpha}_t$ (Sec. 4). (4) $Comp(N_{\mathcal{S}_1}, \dots, N_{\mathcal{S}_T}, N_{\mathcal{T}}, \delta)$ is a function that reflects the convergence behavior, which decreases with larger observation numbers. If we fix $\mathcal{H}, \delta$, $N$ and $N_{\mathcal{T}}$, this term can be viewed as a constant.

**Insights in Representation Learning** Apart from Theorem 1, we propose a novel theoretical analysis in the context of *representation learning*, which motivates practical guidelines in the deep learning regime. We define a stochastic feature function $g$ and we denote its conditional distribution w.r.t. latent variable $Z$ (induced by $g$) as $\mathcal{S}(z|Y=y) = \int_x g(z|x)\mathcal{S}(x|Y=y)dx$. Then we have:

**Theorem 2.** *We assume the settings of loss, the hypothesis are the same with Theorem 1. We further denote the stochastic feature learning function $g : \mathcal{X} \rightarrow \mathcal{Z}$, and the hypothesis $h : \mathcal{Z} \times \mathcal{Y} \rightarrow \mathbb{R}$. Then $\forall \boldsymbol{\lambda}$, the target risk is upper bounded by:*

$$R_{\mathcal{T}}(h, g) \leq \sum_t \boldsymbol{\lambda}[t] R_{\mathcal{S}_t}^{\alpha_t}(h, g) + LK \sum_t \boldsymbol{\lambda}[t] \mathbb{E}_{y \sim \mathcal{T}(y)} W_1(\mathcal{S}_t(z|Y=y) \| \mathcal{T}(z|Y=y)),$$

*where $R_{\mathcal{T}}(h, g) = \mathbb{E}_{(x,y) \sim \mathcal{T}(x,y)} \mathbb{E}_{z \sim g(z|x)} \ell(h(z, y))$.*

Theorem 2 reveal that to control the upper bound, we need to learn $g$ that minimizes the weighted conditional Wasserstein distance and learn $(g, h)$ that minimizes the weighted source risk.

**Comparison with previous Theorems.** Our theory proposed an alternative prospective to understand transfer learning. The first term is $\alpha$-weighted loss. And it will recover the typical source loss minimization if there is no label shift with $\alpha_t(y) \equiv 1$ (Li et al., 2019a; Peng et al., 2019; Zhao et al., 2018; Wen et al., 2020). Beside, minimizing the conditional Wasserstein distances has been shown to be advantageous, compared with $W_1(\mathcal{S}_t(z)\|\mathcal{T}(z))$ (Long et al., 2018). Moreover, Theorem 2 explicitly proposed the theoretical insights about the representation learning function $g$, which remains elusive for previous multi-source transfer theories such as (Wang et al., 2019a; Mansour et al., 2020; Konstantinov & Lampert, 2019; Li et al., 2019a; Peng et al., 2019).

Table 1: Practical principles under different scenarios

| Transfer Scenarios | Learn $(g, h)$ | Estimate $\hat{\alpha}_t$ | Estimate $\boldsymbol{\lambda}$ |
|---|---|---|---|
| Multi-Source Transfer with Limited Target Data | | $\hat{\mathcal{T}}(y)/\hat{\mathcal{S}}_t(y)$ | |
| Unsupervised Multi-Source DA | Sec. 4.1 | Sec. 4.2 | Sec. 4.3 |
| Partial Unsupervised Multi-Source DA | | | |

## 4 UNIFIED PRACTICAL FRAMEWORK IN DEEP LEARNING

The theoretical results in Section 3 motivate general principles to follow when designing multi-source transfer learning algorithms. We summarize those principles in the following rules.

(I) Learn a $g$ that minimizes the weighted conditional Wasserstein distance as well as learn $(g, h)$ that minimizes the $\hat{\alpha}_t$-weighted source risk (Sec. 4.1).

(II) Properly estimate the label distribution ratio $\hat{\alpha}_t$ (Sec. 4.2).

(III) Balance the trade-off between assigning a higher $\boldsymbol{\lambda}[t]$ to the source $t$ that has a smaller weighted prediction loss and conditional Wasserstein distance, and assigning balanced $\boldsymbol{\lambda}[t]$. (Sec. 4.3).

We instantiate these rules with a unified practical framework for solving multi-source transfer learning problems, as shown in Tab 1. We would like to point out that our original theoretical result is based on setting with the available target labels. *The proposed algorithm can be applied to unsupervised scenarios under additional assumptions.*

### 4.1 GUIDELINES IN THE REPRESENTATION LEARNING

Motivated by Theorem 2, given a fixed label ratio estimation $\hat{\alpha}_t$ and fixed $\boldsymbol{\lambda}$, we should find a representation function $g : \mathcal{X} \to \mathcal{Z}$ and a hypothesis function $h : \mathcal{Z} \times \mathcal{Y} \to \mathbb{R}$ such that:

$$\min_{g,h} \sum_t \boldsymbol{\lambda}[t] \hat{R}_{\mathcal{S}_t}^{\hat{\alpha}_t}(h, g) + C_0 \sum_t \boldsymbol{\lambda}[t] \mathbb{E}_{y \sim \hat{\mathcal{T}}(y)} W_1(\hat{\mathcal{S}}_t(z|Y = y) \| \hat{\mathcal{T}}(z|Y = y)) \quad (1)$$

**Explicit Conditional Loss** When target label information is available, one can *explicitly* solve the conditional optimal transport problem with $g$ and $h$ for a given $Y = y$. However, due to the high computational complexity in solving $T \times |\mathcal{Y}|$ optimal transport problems, the original form is practically intractable. To address this issue, we propose to approximate the conditional distribution on latent space $Z$ as Gaussian distribution with identical Covariance matrix such that $\hat{\mathcal{S}}_t(z|Y = y) \approx \mathcal{N}(\mathbf{C}_t^y, \boldsymbol{\Sigma})$ and $\hat{\mathcal{T}}(z|Y = y) \approx \mathcal{N}(\mathbf{C}^y, \boldsymbol{\Sigma})$. Then we have $W_1(\hat{\mathcal{S}}_t(z|Y = y) \| \hat{\mathcal{T}}(z|Y = y)) \leq \|\mathbf{C}_t^y - \mathbf{C}^y\|_2$ (see Appendix G for details). Intuitively, the approximation term is equivalent to the well known *feature mean matching* (Sugiyama & Kawanabe, 2012), which computes the feature centroid of each class (on latent space $Z$) and aligns them by minimizing their $L_2$ distance.

**Implicit Conditional Loss** When target label information is not available (e.g. unsupervised DA and partial DA), the explicit matching approach can adopt pseudo-label predicted by the hypothesis $h$ as a surrogate of the true target label. However, in the early stage of the learning process, the pseudo-labels can be unreliable, which can lead to an inaccurate estimate of $W_1(\hat{\mathcal{S}}(z|Y = y) \| \hat{\mathcal{T}}(z|Y = y))$. To address this, the following Lemma indicates that estimating the conditional Wasserstein distance is equivalent to estimating the Wasserstein adversarial loss weighted by the label-distribution ratio.

**Lemma 1.** *The weighted conditional Wasserstein distance can be implicitly expressed as:*

$$\sum_t \boldsymbol{\lambda}[t] \mathbb{E}_{y \sim \mathcal{T}(y)} W_1(\mathcal{S}_t(z|Y = y) \| \mathcal{T}(z|Y = y)) = \max_{d_1, \cdots, d_T} \sum_t \boldsymbol{\lambda}[t] [\mathbb{E}_{z \sim \mathcal{S}_t(z)} \bar{\alpha}_t(z) d_t(z) - \mathbb{E}_{z \sim \mathcal{T}(z)} d_t(z)],$$

*where $\bar{\alpha}_t(z) = \mathbf{1}_{\{(z,y) \sim \mathcal{S}_t\}} \alpha_t(Y = y)$, and $d_1, \ldots, d_T : \mathcal{Z} \to R_+$ are the 1-Lipschitz domain discriminators (Ganin et al., 2016; Arjovsky et al., 2017).*

Lemma 1 reveals that instead of using pseudo-labels to estimate the weighted conditional Wasserstein distance, one can train $T$ domain discriminators with weighted Wasserstein adversarial loss,

which does not require the pseudo-label of each target sample during the matching. On the other hand, $\bar{\alpha}_t$ can be obtained from $\hat{\alpha}_t$, which will be elaborated in Sec. 3.2.

In practice, we adopt a *hybrid* approach by linearly combining the explicit and implicit matching strategies for all the scenarios, in which empirical results show its effectiveness.

## 4.2 ESTIMATE LABEL DISTRIBUTION RATIO $\hat{\alpha}_t$

**Multi-Source Transfer with target labels** When the target labels are available, $\hat{\alpha}_t$ can be directly estimated from the data without any assumption and $\hat{\alpha}_t \rightarrow \alpha_t$ can be proved from asymptotic statistics.

**Unsupervised Multi-Source DA** In this scenario, it is impossible to estimate a good $\hat{\alpha}_t$ without imposing any additional assumptions. Following (Zhang et al., 2013; Lipton et al., 2018; Azizzadenesheli et al., 2019; Combes et al., 2020), we assume that the conditional distributions are aligned between the target and source domains (i.e., $\mathcal{S}_t(z|y) = \mathcal{T}(z|y)$). Then, we denote $\bar{\mathcal{S}}_t(y), \bar{\mathcal{T}}(y)$ as the predicted $t$-source/target label distribution through the hypothesis $h$, and also define $C_{\hat{\mathcal{S}}_t}[y, k] = \hat{\mathcal{S}}_t[\text{argmax}_{y'} h(z, y') = y, Y = k]$ is the $t$-source *prediction confusion matrix*. We can demonstrate that if the conditional distribution is aligned, we have $\bar{\mathcal{T}}(y) = \bar{\mathcal{T}}_{\hat{\alpha}_t}(y)$, with $\bar{\mathcal{T}}_{\hat{\alpha}_t}(Y = y) = \sum_{k=1}^{\mathcal{Y}} C_{\hat{\mathcal{S}}_t}[y, k]\hat{\alpha}_t(k)$ the constructed target prediction distribution from the $t$-source information. (See Appendix I for the proof). Then we can estimate $\hat{\alpha}_t$ through matching these two distributions by minimizing $D_{\text{KL}}(\bar{\mathcal{T}}(y)\|\bar{\mathcal{T}}_{\hat{\alpha}_t}(y))$, which is equivalent to:

$$\min_{\hat{\alpha}_t} \quad -\sum_{y=1}^{|\mathcal{Y}|} \bar{\mathcal{T}}(y) \log(\sum_{k=1}^{|\mathcal{Y}|} C_{\hat{\mathcal{S}}_t}[y, k]\hat{\alpha}_t(k)) \quad \text{s.t} \quad \forall y \in \mathcal{Y}, \hat{\alpha}_t(y) \geq 0, \quad \sum_{y=1}^{|\mathcal{Y}|} \hat{\alpha}_t(y)\hat{\mathcal{S}}_t(y) = 1 \quad (2)$$

In the aforementioned part, we have assumed the conditional distribution is aligned, which is a feasible requirement in our algorithm, since the goal of $g$ exactly aims at gradually achieving this. In the experiments, we iteratively estimate $\hat{\alpha}_t$ and learn $g$.

**Unsupervised Multi-Source Partial DA** When $\text{supp}(\mathcal{T}(y)) \subseteq \text{supp}(\mathcal{S}_t(y))$, $\alpha_t$ is sparse due to the non-overlapped classes. Accordingly, in addition to the assumption of $\mathcal{S}_t(z|y) = \mathcal{T}(z|y)$ as in unsupervised DA, we also impose such prior knowledge by adding a regularizer $\|\hat{\alpha}_t\|_1$ to the objective of Eq. (2) to induce sparsity in $\hat{\alpha}_t$ (See Appendix J for more details).

In training the neural network, since the non-overlapped classes will be automatically assigned with a small or zero $\hat{\alpha}_t$, $(g, h)$ will be less affected by the classes with small $\hat{\alpha}_t$. Our empirical results effectively validate its capability in detecting non-overlapping classes and show significant improvements over other baselines.

## 4.3 ESTIMATE TASK RELATION COEFFICIENT $\lambda$

Inspired by Theorem 1, given *fixed* $\hat{\alpha}_t$ and $(g, h)$, we estimate $\lambda$ through optimizing the derived upper bound.

$$\min_{\lambda} \quad \sum_t \lambda[t]\hat{R}_{\mathcal{S}_t}^{\hat{\alpha}_t}(h, g) + C_0 \sum_t \lambda[t]\mathbb{E}_{y \sim \hat{\mathcal{T}}(y)} W_1(\hat{\mathcal{T}}(z|Y = y)\|\hat{\mathcal{S}}(z|Y = y)) + C_1 \sqrt{\sum_{t=1}^{T} \frac{\lambda^2[t]}{\beta_t}}$$

$$\text{s.t} \quad \forall t, \lambda[t] \geq 0, \sum_{t=1}^{T} \lambda[t] = 1 \quad (3)$$

In practice, $\hat{R}_{\mathcal{S}_t}^{\hat{\alpha}_t}(h, g)$ is the weighted empirical prediction error and $\mathbb{E}_{y \sim \hat{\mathcal{T}}(y)} W_1(\hat{\mathcal{T}}(z|Y = y)\|\hat{\mathcal{S}}(z|Y = y))$ is approximated by the dynamic feature centroid distance $\sum_y \bar{\mathcal{T}}(y)\|\mathbf{C}_t^y - \mathbf{C}^y\|_2$ (See Appendix L for details). Thus, solving $\lambda$ is a standard convex optimization problem.

## 4.4 ALGORITHM DESCRIPTION

Based on the aforementioned components, we present the description of WADN (Algorithm 1) in the unsupervised scenarios (UDA and Partial DA), which iteratively updates $(g, h)$, $\hat{\alpha}_t$, and $\lambda$. When

---

**Algorithm 1** Wasserstein Aggregation Domain Network (unsupervised scenarios, one iteration)

---

**Require:** Labeled source samples $\hat{\mathcal{S}}_1, \ldots, \hat{\mathcal{S}}_T$, Target samples $\hat{\mathcal{T}}$
**Ensure:** Label distribution ratio $\hat{\alpha}_t$, task relation simplex $\boldsymbol{\lambda}$. Feature Function $g$, Classifier $h$, Domain critic function $d_1, \ldots, d_T$, class centroid for source $\mathbf{C}_t^y$ and target $\mathbf{C}^y$ ($\forall t = [1, T], y \in \mathcal{Y}$).

1: ▷▷▷ DNN Parameter Training Stage (fixed $\alpha_t$ and $\boldsymbol{\lambda}$) ◁◁◁
2: **for** mini-batch of samples $(\mathbf{x}_{\mathcal{S}_1}, \mathbf{y}_{\mathcal{S}_1}) \sim \hat{\mathcal{S}}_1, \ldots, (\mathbf{x}_{\mathcal{S}_T}, \mathbf{y}_{\mathcal{S}_T}) \sim \hat{\mathcal{S}}_T, (\mathbf{x}_{\mathcal{T}}) \sim \hat{\mathcal{T}}$ **do**
3:     Predict target pseudo-label $\bar{\mathbf{y}}_{\mathcal{T}} = \text{argmax}_y h(g(\mathbf{x}_{\mathcal{T}}), y)$
4:     Compute source confusion matrix for each batch (un-normalized)
        $C_{\hat{\mathcal{S}}_t} = \#[\text{argmax}_{y'} h(z, y') = y, Y = k]$ $(t = 1, \ldots, T)$
5:     Compute the *batched* class centroid for source $C_t^y$ and target $C^y$.
6:     Moving Average for update source/target class centroid: ($\epsilon_1 = 0.7$)
7:         Update Source class centroid    $\mathbf{C}_t^y = \epsilon_1 \times \mathbf{C}_t^y + (1 - \epsilon_1) \times C_t^y$
8:         Update Target class centroid    $\mathbf{C}^y = \epsilon_1 \times \mathbf{C}^y + (1 - \epsilon_1) \times C^y$
9:     Updating $g, h, d_1, \ldots, d_T$ (SGD and Gradient Reversal), by solving:

$$\min_{g,h} \max_{d_1, \ldots, d_T} \underbrace{\sum_t \boldsymbol{\lambda}[t] \hat{R}_{\mathcal{S}_t}^{\hat{\alpha}_t}(h, g)}_{\text{Classification Loss}} + \epsilon C_0 \underbrace{\sum_t \boldsymbol{\lambda}[t] \mathbb{E}_{y \sim \bar{\mathcal{T}}(y)} \|\mathbf{C}_t^y - \mathbf{C}^y\|_2}_{\text{Explicit Conditional Loss}}$$

$$+ (1 - \epsilon) C_0 \underbrace{\sum_t \boldsymbol{\lambda}[t] [\mathbb{E}_{z \sim \hat{\mathcal{S}}_t(z)} \bar{\alpha}_t(z) d(z) - \mathbb{E}_{z \sim \hat{\mathcal{T}}(z)} d(z)]}_{\text{Implicit Conditional Loss}}$$

10: **end for**
11: ▷▷▷ Estimation $\hat{\alpha}_t$ and $\boldsymbol{\lambda}$ ◁◁◁
12: Compute the global(normalized) source confusion matrix
    $C_{\hat{\mathcal{S}}_t} = \hat{\mathcal{S}}_t[\text{argmax}_{y'} h(z, y') = y, Y = k]$ $(t = 1, \ldots, T)$
13: Solve $\alpha_t$ (denoted as $\{\alpha_t'\}_{t=1}^T$) by (Sec. 4.2 Unsupervised DA or Partial UDA).
14: Update $\alpha_t$ by moving average: $\alpha_t = \epsilon_1 \times \alpha_t + (1 - \epsilon_1) \times \alpha_t'$
15: Compute the weighted loss and weighted centroid distance, then solve $\boldsymbol{\lambda}$ (denoted as $\boldsymbol{\lambda}'$) from Sec. 4.3. And updating $\boldsymbol{\lambda}$ by moving average: $\boldsymbol{\lambda} = \epsilon_1 \times \boldsymbol{\lambda} + (1 - \epsilon_1) \times \boldsymbol{\lambda}'$

---

updating $\boldsymbol{\lambda}$ and $\alpha_t$, we used package CVXPY to optimize the two standard convex losses after each training epoch, then we updating them by using the moving average. As for WADN under target label information, we did not require pseudo-label and directly compute $\hat{\alpha}_t$, shown in Appendix L.

## 5 EXPERIMENTS

In this section, we compare proposed approaches with several baselines for the popular tasks. For all the scenarios, the following baselines are evaluated: (I) **Source** method applied only labelled source data to train the model. (II) **DANN** (Ganin et al., 2016). We follow the protocol of Wen et al. (2020) to merge all the source dataset as a global source domain. (III) **MDAN** (Zhao et al., 2018); (IV) **MDMN** (Li et al., 2018b); (V) **M³SDA** (Peng et al., 2019) adopted maximizing classifier discrepancy (Saito et al., 2018) and (VI) **DARN** (Wen et al., 2020). For the conventional multi-source transfer and partial unsupervised multi-source DA, we additionally compare specific baselines. All the baselines are re-implemented in the same network structure for fair comparisons. The detailed network structures, hyper-parameter settings, training details are put in Appendix M.

We evaluate the performance on three different datasets: (I) **Amazon Review.** (Blitzer et al., 2007) It contains four domains (Books, DVD, Electronics, and Kitchen) with positive and negative product reviews. We follow the common data pre-processing strategies as (Chen et al. (2012)) to form a 5000-dimensional bag-of-words feature. Note that the label distribution in the original dataset is uniform. *To enhance the benefits of the proposed approach, we create a label distribution drifted task by randomly dropping* 50% *negative reviews of all the sources while keeping the target identical.* (show in Fig.3 (a)). (II) **Digits**. It consists four digits recognition datasets including MNIST, USPS (Hull, 1994), SVHN (Netzer et al., 2011) and Synth (Ganin et al., 2016). *We also create a slight label distribution drift for the sources by randomly dropping* 50% *samples on digits 5-9 and keep*

*target identical.* (showed in Fig.(3)(b)). (III) **Office-Home Dataset** (Venkateswara et al., 2017). It contains 65 classes for four different domains: Art, Clipart, Product and Real-World. We used the ResNet50 (He et al., 2016) pretrained from the ImageNet in PyTorch as the base network for feature learning and put a MLP for the classification. The label distributions in these four domains are different and we did not manually create a label drift (showed in Fig.3 (c)).

## 5.1 UNSUPERVISED MULTI-SOURCE DA

In the unsupervised multi-source DA, we evaluate the proposed approach on all the three datasets. We use a similar hyper-parameter selection strategy as in DANN (Ganin et al., 2016). All reported results are averaged from five runs. The detailed experimental settings are illustrated in Appendix M. The empirical results are illustrated in Tab. 7, 2 and 3. Since we did not change the target label distribution throughout the whole experiments, then we still use the target accuracy as the metric. We report the means and standard deviations for each approach. The best approaches based on a two-sided Wilcoxon signed-rank test (significance level $p = 0.05$) are shown in bold.

Table 2: Unsupervised DA: Accuracy (%) on the Source-Shifted Digits.

| Target | MNIST | SVHN | SYNTH | USPS | Average |
|---|---|---|---|---|---|
| Source | $84.93_{\pm 1.50}$ | $67.14_{\pm 1.40}$ | $78.11_{\pm 1.31}$ | $86.02_{\pm 1.12}$ | 79.05 |
| DANN | $86.99_{\pm 1.53}$ | $69.56_{\pm 2.26}$ | $78.73_{\pm 1.30}$ | $86.81_{\pm 1.74}$ | 80.52 |
| MDAN | $87.86_{\pm 2.24}$ | $69.13_{\pm 1.56}$ | $79.77_{\pm 1.69}$ | $86.50_{\pm 1.59}$ | 80.81 |
| MDMN | $87.31_{\pm 1.88}$ | $69.84_{\pm 1.59}$ | $80.27_{\pm 0.88}$ | $86.61_{\pm 1.41}$ | 81.00 |
| M$^3$SDA | $87.22_{\pm 1.70}$ | $68.89_{\pm 1.93}$ | $80.01_{\pm 1.77}$ | $86.39_{\pm 1.68}$ | 80.87 |
| DARN | $86.98_{\pm 1.29}$ | $68.59_{\pm 1.79}$ | $80.68_{\pm 0.61}$ | $86.85_{\pm 1.78}$ | 80.78 |
| WADN | $\mathbf{89.07}_{\pm 0.72}$ | $\mathbf{71.66}_{\pm 0.77}$ | $\mathbf{82.06}_{\pm 0.89}$ | $\mathbf{90.07}_{\pm 1.10}$ | **83.22** |
| Full TAR | $98.70_{\pm 0.15}$ | $85.20_{\pm 0.09}$ | $95.10_{\pm 0.14}$ | $96.64_{\pm 0.13}$ | 93.91 |

Table 3: Unsupervised DA: Accuracy (%) on Office-Home

| Target | Art | Clipart | Product | Real-World | Average |
|---|---|---|---|---|---|
| Source | $49.25_{\pm 0.60}$ | $46.89_{\pm 0.61}$ | $66.54_{\pm 1.72}$ | $73.64_{\pm 0.91}$ | 59.08 |
| DANN | $50.32_{\pm 0.32}$ | $50.11_{\pm 1.16}$ | $68.18_{\pm 1.27}$ | $73.71_{\pm 1.63}$ | 60.58 |
| MDAN | $67.93_{\pm 0.36}$ | $66.61_{\pm 1.32}$ | $79.24_{\pm 1.52}$ | $81.82_{\pm 0.65}$ | 73.90 |
| MDMN | $68.38_{\pm 0.58}$ | $67.42_{\pm 0.53}$ | $82.49_{\pm 0.56}$ | $83.32_{\pm 1.93}$ | 75.28 |
| M$^3$SDA | $63.77_{\pm 1.07}$ | $62.30_{\pm 0.44}$ | $75.85_{\pm 1.24}$ | $79.92_{\pm 0.60}$ | 70.46 |
| DARN | $69.89_{\pm 0.42}$ | $68.61_{\pm 0.50}$ | $83.37_{\pm 0.62}$ | $84.29_{\pm 0.46}$ | 76.54 |
| WADN | $\mathbf{73.78}_{\pm 0.43}$ | $\mathbf{70.18}_{\pm 0.54}$ | $\mathbf{86.32}_{\pm 0.38}$ | $\mathbf{87.28}_{\pm 0.87}$ | **79.39** |
| Full TAR | $76.17_{\pm 0.16}$ | $79.37_{\pm 0.22}$ | $90.60_{\pm 0.24}$ | $87.65_{\pm 0.18}$ | 83.45 |

The empirical results reveal a significantly better performance ($\approx 3\%$) on different datasets. For understanding the working principles of WADN, we evaluate the performance under different levels of source label shift in Amazon Review dataset (Fig.1(a)). The results show strong practical benefits for WADN during a gradual larger label shift. In addition, we visualize the task relations in digits (Fig.1(b)) and demonstrate a *non-uniform* $\boldsymbol{\lambda}$, which highlights the importance of properly choosing the most related source rather than simply merging all the data. E.g. when the target domain is SVHN, WADN mainly leverages the information from SYNTH, since they are more semantically similar and MNIST does not help too much for SVHN (observed by Ganin et al. (2016)). The additional analysis and results can be found in Appendix O.

## 5.2 MULTI-SOURCE TRANSFER LEARNING WITH LIMITED TARGET SAMPLES

We adopt Amazon Review and Digits in the multi-source transfer learning with limited target samples, which have been widely used. In the experiments, we still use shifted sources. We randomly sample only 10% labeled samples (w.r.t. target dataset in unsupervised DA) as training set and the rest 90% samples as the unseen target test set. (See Appendix M for details). We adopt the same

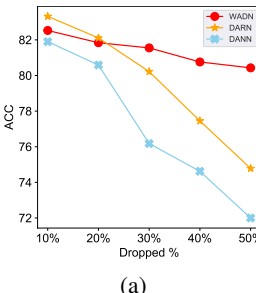 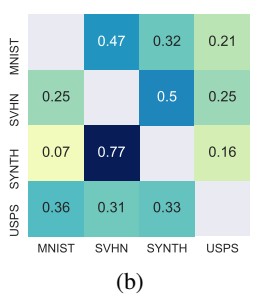 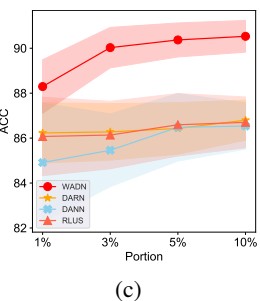

(a)          (b)          (c)

Figure 1: (a) Unsupervised DA with Amazon Review dataset. Accuracy under different levels of shifted sources (higher dropping rate means larger label shift). The results are averaged on all target domains. See the results for each task in Fig. (12). (b) Visualization of $\lambda$ in unsupervised DA, each row corresponds to one target domain. (c) Transfer learning with limited target labels in USPS. The performance of WADN is consistently better under different target samples (smaller portion indicates fewer target samples).

hyper-parameters and training strategies with unsupervised DA. We specifically add a recent baseline RLUS (Konstantinov & Lampert, 2019) and MME (Saito et al., 2019), which also considered transfer learning with the labeled target.

Table 4: Multi-source Transfer: Accuracy (%) on Source-Shifted Amazon Review

| Target | Books | DVD | Electronics | Kitchen | Average |
|---|---|---|---|---|---|
| Source + Tar | $72.59_{\pm1.89}$ | $73.02_{\pm1.84}$ | $81.59_{\pm1.58}$ | $77.03_{\pm1.73}$ | 76.06 |
| DANN | $67.35_{\pm2.28}$ | $66.33_{\pm2.42}$ | $78.03_{\pm1.72}$ | $74.31_{\pm1.71}$ | 71.50 |
| MDAN | $68.70_{\pm2.99}$ | $69.30_{\pm2.21}$ | $78.78_{\pm2.21}$ | $74.07_{\pm1.89}$ | 72.71 |
| $M^3SDA$ | $69.28_{\pm1.78}$ | $67.40_{\pm0.46}$ | $76.28_{\pm0.81}$ | $76.50_{\pm1.19}$ | 72.36 |
| DARN | $68.57_{\pm1.35}$ | $68.77_{\pm1.81}$ | $80.19_{\pm1.66}$ | $77.51_{\pm1.20}$ | 73.76 |
| RLUS | $71.83_{\pm1.71}$ | $69.64_{\pm2.39}$ | $81.98_{\pm1.04}$ | $78.69_{\pm1.15}$ | 75.54 |
| MME | $69.66_{\pm0.58}$ | $71.36_{\pm0.96}$ | $78.88_{\pm1.51}$ | $76.64_{\pm1.73}$ | 74.14 |
| WADN | $\mathbf{74.83}_{\pm0.84}$ | $\mathbf{75.05}_{\pm0.62}$ | $\mathbf{84.23}_{\pm0.58}$ | $\mathbf{81.53}_{\pm0.90}$ | $\mathbf{78.91}$ |
| Full TAR | $84.10_{\pm0.13}$ | $83.68_{\pm0.12}$ | $86.11_{\pm0.32}$ | $88.72_{\pm0.14}$ | 86.65 |

Table 5: Multi-source Transfer: Accuracy (%) on the Source-Shifted Digits

| Target | MNIST | SVHN | SYNTH | USPS | Average |
|---|---|---|---|---|---|
| Source + Tar | $79.63_{\pm1.74}$ | $56.48_{\pm1.90}$ | $69.64_{\pm1.38}$ | $86.29_{\pm1.56}$ | 73.01 |
| DANN | $86.77_{\pm1.30}$ | $69.13_{\pm1.09}$ | $78.82_{\pm1.35}$ | $86.54_{\pm1.03}$ | 80.32 |
| MDAN | $86.93_{\pm1.05}$ | $68.25_{\pm1.53}$ | $79.80_{\pm1.17}$ | $86.23_{\pm1.41}$ | 80.30 |
| $M^3SDA$ | $85.88_{\pm2.06}$ | $68.84_{\pm1.05}$ | $76.29_{\pm0.95}$ | $87.15_{\pm1.10}$ | 79.54 |
| DARN | $86.58_{\pm1.46}$ | $68.86_{\pm1.30}$ | $80.47_{\pm0.67}$ | $86.80_{\pm0.89}$ | 80.68 |
| RLUS | $87.61_{\pm1.08}$ | $\mathbf{70.50}_{\pm0.94}$ | $79.52_{\pm1.30}$ | $86.70_{\pm1.13}$ | 81.08 |
| MME | $87.24_{\pm0.95}$ | $65.20_{\pm1.35}$ | $80.31_{\pm0.60}$ | $87.88_{\pm0.76}$ | 80.16 |
| WADN | $\mathbf{88.32}_{\pm1.17}$ | $\mathbf{70.64}_{\pm1.02}$ | $\mathbf{81.53}_{\pm1.11}$ | $\mathbf{90.53}_{\pm0.71}$ | $\mathbf{82.75}$ |
| Full TAR | $98.70_{\pm0.15}$ | $85.20_{\pm0.09}$ | $95.10_{\pm0.14}$ | $96.64_{\pm0.13}$ | 93.91 |

The results are reported in Tabs. 4, 5, which also indicates strong empirical benefits. To show the effectiveness of WADN, we select various portions of labelled samples ($1\% \sim 10\%$) on the target. The results in Fig.1(c) on USPS dataset shows a consistently better than the baselines, even in the few target samples.

### 5.3 PARTIAL UNSUPERVISED MULTI-SOURCE DA

In this scenario, we adopt the Office-Home dataset to evaluate our approach, as it contains large (65) classes. We do not change the source domains and we randomly choose 35 classes from the target. We evaluate all the baselines on the same selected classes and repeat 5 times. All reported results are averaged from 3 different sub-class selections (15 runs in total), showing in Tab.6. (See Appendix M for details.) We additionally compare PADA (Cao et al., 2018) approach by merging all the sources and use *one-to-one* partial DA algorithm. We adopt the same hyper-parameters and training strategies with unsupervised DA scenario.

Table 6: Unsupervised Partial DA: Accuracy (%) on Office-Home (#Source: 65, #Target: 35)

| Target | Art | Clipart | Product | Real-World | Average |
|---|---|---|---|---|---|
| Source | $50.56_{\pm1.42}$ | $49.79_{\pm1.14}$ | $68.10_{\pm1.33}$ | $78.24_{\pm0.76}$ | 61.67 |
| DANN | $53.86_{\pm2.23}$ | $52.71_{\pm2.20}$ | $71.25_{\pm2.44}$ | $76.92_{\pm1.21}$ | 63.69 |
| MDAN | $67.56_{\pm1.39}$ | $65.38_{\pm1.30}$ | $81.49_{\pm1.92}$ | $83.44_{\pm1.01}$ | 74.47 |
| MDMN | $68.13_{\pm1.08}$ | $65.27_{\pm1.93}$ | $81.33_{\pm1.29}$ | $84.00_{\pm0.64}$ | 74.68 |
| M³SDA | $65.10_{\pm1.97}$ | $61.80_{\pm1.99}$ | $76.19_{\pm2.44}$ | $79.14_{\pm1.51}$ | 70.56 |
| DARN | $71.53_{\pm0.63}$ | $69.31_{\pm1.08}$ | $82.87_{\pm1.56}$ | $84.76_{\pm0.57}$ | 77.12 |
| PADA | $74.37_{\pm0.84}$ | $69.64_{\pm0.80}$ | $83.45_{\pm1.13}$ | $85.64_{\pm0.39}$ | 78.28 |
| WADN | $\mathbf{80.06}_{\pm0.93}$ | $\mathbf{75.90}_{\pm1.06}$ | $\mathbf{89.55}_{\pm0.72}$ | $\mathbf{90.40}_{\pm0.39}$ | **83.98** |

The reported results are also significantly better than the current multi-source DA or one-to-one partial DA approach, which verifies the benefits of WADN: properly estimating $\hat{\alpha}_t$ and assigning proper $\lambda$ for each source.

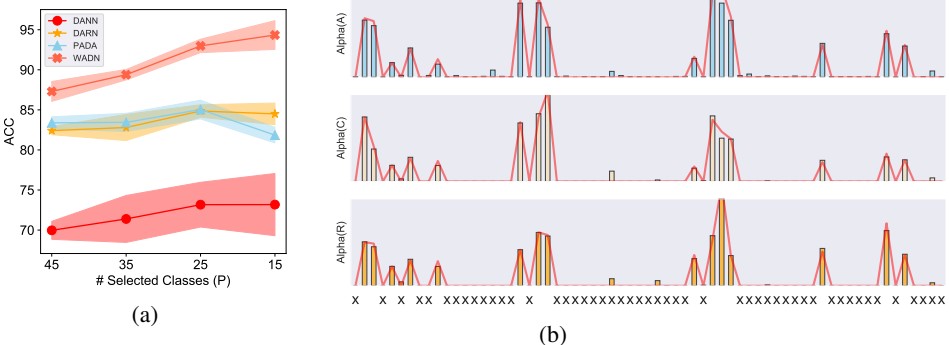

(a)

(b)

Figure 2: Analysis on Partial DA of target Product. (a) Performance (mean $\pm$ std) of different selected classes on the target; (b) We select 15 classes and visualize estimated $\hat{\alpha}_t$ (the bar plot). The "X" along the x-axis represents the index of *dropped* 50 classes. The red curves are the true label distribution ratio. See Appendix P for additional results and analysis.

Besides, we change the number of selected classes (Fig 2(a)), the proposed WADN still indicates consistent better results by a large margin, which indicates the importance of considering $\hat{\alpha}_t$ and $\lambda$. In contrast, DANN shows unstable results in less selected classes. (See Appendix P for details) Beside, WADN shows a good estimation of the label distribution ratio (Fig 2(b)) and has correctly detected the non-overlapping classes, which indicates its good explainability.

## 6 CONCLUSION

In this paper, we proposed a new theoretical principled algorithm WADN (Wasserstein Aggregation Domain Network) to solve the multi-source transfer learning problem under target shift. WADN provides a unified solution for various deep multi-source transfer scenarios: learning with limited target data, unsupervised DA, and partial unsupervised DA. We evaluate the proposed method by extensive experiments and show its strong empirical results.

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

## A  ADDITIONAL EMPIRICAL RESULTS

Table 7: Unsupervised DA: Accuracy $(\%)$ on Source-Shifted Amazon Review.

| Target | Books | DVD | Electronics | Kitchen | Average |
|---|---|---|---|---|---|
| Source | $68.15_{\pm 1.37}$ | $69.51_{\pm 0.74}$ | $82.09_{\pm 0.88}$ | $75.30_{\pm 1.29}$ | 73.81 |
| DANN | $65.59_{\pm 1.35}$ | $67.23_{\pm 0.71}$ | $80.49_{\pm 1.11}$ | $74.71_{\pm 1.53}$ | 72.00 |
| MDAN | $68.77_{\pm 2.31}$ | $67.81_{\pm 2.46}$ | $80.96_{\pm 0.77}$ | $75.67_{\pm 1.96}$ | 73.30 |
| MDMN | $70.56_{\pm 1.05}$ | $69.64_{\pm 0.73}$ | $82.71_{\pm 0.71}$ | $77.05_{\pm 0.78}$ | 74.99 |
| $\text{M}^3\text{SDA}$ | $69.09_{\pm 1.26}$ | $68.67_{\pm 1.37}$ | $81.34_{\pm 0.66}$ | $76.10_{\pm 1.47}$ | 73.79 |
| DARN | $71.21_{\pm 1.16}$ | $68.68_{\pm 1.12}$ | $81.51_{\pm 0.81}$ | $77.71_{\pm 1.09}$ | 74.78 |
| WADN | $\mathbf{73.72}_{\pm 0.63}$ | $\mathbf{79.64}_{\pm 0.34}$ | $\mathbf{84.64}_{\pm 0.48}$ | $\mathbf{83.73}_{\pm 0.50}$ | **80.43** |
| Full TAR | $84.10_{\pm 0.13}$ | $83.68_{\pm 0.12}$ | $86.11_{\pm 0.32}$ | $88.72_{\pm 0.14}$ | 86.65 |

## B  ADDITIONAL RELATED WORK

**Multi-source transfer learning Practice** has been proposed from various prospective. The key idea is to estimate the importance of different sources and then select the most related ones, to mitigate the influence of negative transfer. In the multi-source unsupervised DA, (Sankaranarayanan et al., 2018; Balaji et al., 2019; Pei et al., 2018; Zhao et al., 2019; Zhu et al., 2019; Zhao et al., 2020; 2019; Stojanov et al., 2019; Li et al., 2019b; Wang et al., 2019b; Lin et al., 2020) proposed different practical strategies in the classification, regression and semantic segmentation problems. In the presence of target labels, Hoffman et al. (2012); Tan et al. (2013); Wei et al. (2017); Yao & Doretto (2010); Konstantinov & Lampert (2019) used generalized linear model to learn the target. Christodoulidis et al. (2016); Li et al. (2019a); Chen et al. (2019) focused on deep learning approaches and Lee et al. (2019) proposed an ad-hoc strategy to combine to sources in the few-shot target domains. These ideas are generally data-driven approaches and do not analyze the why the proposed practice can control the generalization error.

**Label-Partial Transfer Learning** Label-Partial can be viewed as a special case of the label-shift. [1] Most existing works focus on one-to-one partial transfer learning (Zhang et al., 2018; Chen et al., 2020; Bucci et al., 2019; Cao et al., 2019) by adopting the re-weighting training approach without a formal understanding. In our paper, we first rigorously analyzed this common practice and adopt the label distribution ratio as its weights, which provides a principled approach in this scenario.

### B.1  OTHER SCENARIOS RELATED TO MULTI-SOURCE TRANSFER LEARNING

**Domain Generalization** The domain generalization (DG) resembles multi-source transfer but aims at different goals. A common setting in DG is to learn multiple source but directly predict on the unseen target domain. The conventional DG approaches generally learn a distribution invariant features (Balaji et al., 2018; Saenko et al., 2010; Motiian et al., 2017; Ilse et al., 2019) or conditional distribution invariant features (Li et al., 2018a; Akuzawa et al., 2019). However, our theoretical results reveal that in the presence of label shift (i.e $\alpha_t(y) \neq 1$) and outlier tasks then learning conditional or marginal invariant features can not guarantee a small target risk. Our theoretical result enables a formal understanding about the inherent difficulty in DG problems.

**Few-Shot Learning** The few-shot learning (Finn et al., 2017; Snell et al., 2017; Sung et al., 2018) can be viewed as a very specific scenario of multi-source transfer learning. We would like to point out the differences between the few-shot learning and our paper. (1) Few-shot learning generally involves a **very large set** of source domains $T \gg 1$ and each domain consists a **modest number** of observations $N_{\mathcal{S}_t}$. In our paper, we are interested in the a **modest number** of source domains $T$ but each source domain including a **sufficient large** number of observations ($N_{\mathcal{S}_t} \gg 1$). (2) In the target domain, the few-shot setting generally used K-samples ($K$ is very small) for each class for the fine-tuning. We would like to point out this setting generally violates our theoretical assumption. In

---

[1] Since $\text{supp}(\mathcal{T}(y)) \subseteq \text{supp}(\mathcal{S}_t(y))$ then we naturally have $\mathcal{T}(y) \neq \mathcal{S}_t(y)$.

our paper, we assume the target data is i.i.d. sampled from $\mathcal{D}(x, y)$. It is equivalently viewed that we first i.i.d. sample $y \sim \mathcal{D}(y)$, then i.i.d. sample $x \sim \mathcal{D}(x|y)$. Generally the $\mathcal{D}(y)$ is **non-uniform**, thus few-shot setting are generally not applicable for our theoretical assumptions.

**Multi-Task Learning** The goal of multi-task learning (Zhang & Yang, 2017) aims to improve the prediction performance of **all** the tasks. In our paper, we aim at controlling the prediction risk of a specified target domain. We also notice some practical techniques are common such as the shared parameter (Zhang & Yeung, 2012), shared representation (Ruder, 2017), etc.

## C  ADDITIONAL FIGURES RELATED TO THE MAIN PAPER

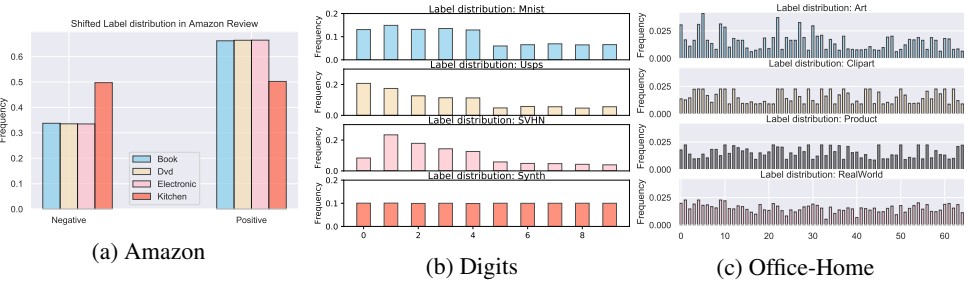

(a) Amazon  (b) Digits  (c) Office-Home

Figure 3: Label distribution visualization. (a) One example in Amazon Review dataset with sources: Book, Dvd, Electronic and target: Kitchen. We randomly drop 50% of the negative reviews in all the sources while keeping target label distribution unchanged. (b) One example in Digits dataset with Sources: MNIST, USPS, SVHN and Target Synth. We randomly drop 50% data on digits 5-9 in all sources while keeping target label distribution unchanged. (c) Office-Home dataset. The original label distribution is non-uniform. See Appendix M for details.

## D  TABLE OF NOTATION

Table 8: Table of Notations

| | |
|---|---|
| $R_{\mathcal{D}}(h) = \mathbb{E}_{(x,y)\sim\mathcal{D}}\ell(h(x,y))$ | Expected Risk on distribution $\mathcal{D}$ w.r.t. hypothesis $h$ |
| $\hat{R}_{\mathcal{D}}(h) = \frac{1}{N}\sum_{i=1}^{N}\ell(h(x_i, y_i))$ | Empirical Risk on observed data $\{(x_i, y_i)\}_{i=1}^{N}$ that are i.i.d. sampled from $\mathcal{D}$. |
| $\alpha$ and $\hat{\alpha}_t$ | True and empirical label distribution ratio $\alpha(y) = \mathcal{T}(y)/\mathcal{S}(y)$ |
| $\hat{R}_{\mathcal{S}}^{\alpha}(h) = \frac{1}{N}\sum_{i=1}^{N}\alpha(y_i)\ell(h(x_i, y_i))$ | Empirical Weighted Risk on observed data $\{(x_i, y_i)\}_{i=1}^{N}$. |
| $\mathcal{S}(z|y) = \int_x g(z|x)S(x|Y=y)dx$ | Conditional distribution w.r.t. latent variable $Z$ that induced by feature learning function $g$. |
| $W_1(\mathcal{S}_t(z|y)\|\mathcal{T}(z|y))$ | Conditional Wasserstein distance on the latent space $Z$ |

## E  PROOF OF THEOREM 1

**Proof idea** Theorem 1 consists three steps in the proof:

**Lemma 2.** *If the prediction loss is assumed as $L$-Lipschitz and the hypothesis is $K$-Lipschitz w.r.t. the feature $x$ (given the same label), i.e. for $\forall Y = y$, $\|h(x_1, y) - h(x_2, y)\|_2 \leq K\|x_1 - x_2\|_2$. Then the target risk can be upper bounded by:*

$$R_{\mathcal{T}}(h) \leq \sum_t \boldsymbol{\lambda}[t]R_{\mathcal{S}}^{\alpha_t}(h) + LK\sum_t \boldsymbol{\lambda}[t]\mathbb{E}_{y\sim\mathcal{T}(y)}W_1(\mathcal{T}(x|Y=y)\|\mathcal{S}(x|Y=y)) \quad (4)$$

*Proof.* The target risk can be expressed as:

$$R_{\mathcal{T}}(h(x,y)) = \mathbb{E}_{(x,y)\sim\mathcal{T}}\ell(h(x,y)) = \mathbb{E}_{y\sim\mathcal{T}(y)}\mathbb{E}_{x\sim\mathcal{T}(x|y)}\ell(h(x,y))$$

By denoting $\alpha(y) = \frac{\mathcal{T}(y)}{\mathcal{S}(y)}$, then we have:

$$\mathbb{E}_{y\sim\mathcal{T}(y)}\mathbb{E}_{y\sim\mathcal{T}(x|y)}\ell(h(x,y)) = \mathbb{E}_{y\sim\mathcal{S}(y)}\alpha(y)\mathbb{E}_{x\sim\mathcal{T}(x|y)}\ell(h(x,y))$$

Then we aim to upper bound $\mathbb{E}_{x\sim\mathcal{T}(x|y)}\ell(h(x,y))$. For any fixed $y$,

$$\mathbb{E}_{x\sim\mathcal{T}(x|y)}\ell(h(x,y)) - \mathbb{E}_{x\sim\mathcal{S}(x|y)}\ell(h(x,y)) \leq |\int_{x\in\mathcal{X}}\ell(h(x,y))d(\mathcal{T}(x|y)-\mathcal{S}(x|y))|$$

Then according to the Kantorovich-Rubinstein duality, for **any** distribution coupling $\gamma \in \Pi(\mathcal{T}(x|y),\mathcal{S}(x|y))$, then we have:

$$= \inf_{\gamma} |\int_{\mathcal{X}\times\mathcal{X}}\ell(h(x_p,y)) - \ell(h(x_q,y))d\gamma(x_p,x_q)|$$
$$\leq \inf_{\gamma} \int_{\mathcal{X}\times\mathcal{X}}|\ell(h(x_p,y)) - \ell(h(x_q,y))|d\gamma(x_p,x_q)$$
$$\leq L\inf_{\gamma} \int_{\mathcal{X}\times\mathcal{X}}|h(x_p,y)) - h(x_q,y)|d\gamma(x_p,x_q)$$
$$\leq LK\inf_{\gamma} \int_{\mathcal{X}\times\mathcal{X}}\|x_p - x_q\|_2 d\gamma(x_p,x_q)$$
$$= LKW_1(\mathcal{T}(x|Y=y)\|\mathcal{S}(x|Y=y))$$

The first inequality is obvious; and the second inequality comes from the assumption that $\ell$ is $L$-Lipschitz; the third inequality comes from the hypothesis is $K$-Lipschitz w.r.t. the feature $x$ (given the same label), i.e. for $\forall Y=y$, $\|h(x_1,y)-h(x_2,y)\|_2 \leq K\|x_1 - x_2\|_2$.

Then we have:

$$R_\mathcal{T}(h) \leq \mathbb{E}_{y\sim\mathcal{S}(y)}\alpha(y)[\mathbb{E}_{x\sim\mathcal{S}(x|y)}\ell(h(x,y)) + LKW_1(\mathcal{T}(x|y)\|\mathcal{S}(x|y))]$$
$$= \mathbb{E}_{(x,y)\sim\mathcal{S}}\alpha(y)\ell(h(x,y)) + LK\mathbb{E}_{y\sim\mathcal{T}(y)}W_1(\mathcal{T}(x|Y=y)\|\mathcal{S}(x|Y=y))$$
$$= R_\mathcal{S}^\alpha(h) + LK\mathbb{E}_{y\sim\mathcal{T}(y)}W_1(\mathcal{T}(x|Y=y)\|\mathcal{S}(x|Y=y))$$

Supposing each source $\mathcal{S}_t$ we assign the weight $\boldsymbol{\lambda}[t]$ and label distribution ratio $\alpha_t(y) = \frac{\mathcal{T}(y)}{\mathcal{S}_t(y)}$, then by combining this $T$ source target pair, we have:

$$R_\mathcal{T}(h) \leq \sum_t \boldsymbol{\lambda}[t]R_{\mathcal{S}_t}^{\alpha_t}(h) + LK\sum_t \boldsymbol{\lambda}[t]\mathbb{E}_{y\sim\mathcal{T}(y)}W_1(\mathcal{T}(x|Y=y)\|\mathcal{S}_t(x|Y=y))$$

$\square$

Then we will prove Theorem 1 from this result, we will derive the non-asymptotic bound, estimated from the finite sample observations. Supposing the empirical label ratio value is $\hat{\alpha}_t$, then for any simplex $\boldsymbol{\lambda}$ we can prove the high-probability bound.

### E.1 BOUNDING THE EMPIRICAL AND EXPECTED PREDICTION RISK

*Proof.* We first bound the first term, which can be upper bounded as:

$$\sup_h |\sum_t \boldsymbol{\lambda}[t]R_{\mathcal{S}_t}^{\alpha_t}(h) - \sum_t \boldsymbol{\lambda}[t]\hat{R}_{\mathcal{S}_t}^{\hat{\alpha}_t}(h)| \leq \underbrace{\sup_h |\sum_t \boldsymbol{\lambda}[t]R_{\mathcal{S}_t}^{\alpha_t}(h) - \sum_t \boldsymbol{\lambda}[t]\hat{R}_{\mathcal{S}_t}^{\alpha_t}(h)|}_{(I)} + \underbrace{\sup_h |\sum_t \boldsymbol{\lambda}[t]\hat{R}_{\mathcal{S}_t}^{\alpha_t}(h) - \sum_t \boldsymbol{\lambda}[t]\hat{R}_{\mathcal{S}_t}^{\hat{\alpha}_t}(h)|}_{(II)}$$

**Bounding term (I)** According to the McDiarmid inequality, each item changes at most $|\frac{2\boldsymbol{\lambda}[t]\alpha_t(y)\ell}{N_{\mathcal{S}_t}}|$. Then we have:

$$P((I) - \mathbb{E}(I) \geq t) \leq \exp(\frac{-2t^2}{\sum_{t=1}^T \frac{4}{\beta_t N}\boldsymbol{\lambda}^2[t]\alpha_t(y)^2\ell^2}) = \delta$$

By substituting $\delta$, at high probability $1 - \delta$ we have:

$$(\text{I}) \leq \mathbb{E}(\text{I}) + L_{\max} d_\infty^{\sup} \sqrt{\sum_{t=1}^{T} \frac{\boldsymbol{\lambda}[t]^2}{\beta_t}} \sqrt{\frac{\log(1/\delta)}{2N}}$$

Where $L_{\max} = \sup_{h \in \mathcal{H}} \ell(h)$ and $N = \sum_{t=1}^{T} N_{\mathcal{S}_t}$ the total source observations and $\beta_t = \frac{N_{\mathcal{S}_t}}{N}$ the frequency ratio of each source. And $d_\infty^{\sup} = \max_{t=1,\ldots,T} d_\infty(\mathcal{T}(y) \| \mathcal{S}(y)) = \max_{t=1,\ldots,T} \max_{y \in [1, \mathcal{Y}]} \alpha_t(y)$, the maximum true label shift value (constant).

Bounding $\mathbb{E} \sup(\text{I})$, the expectation term can be upper bounded as the form of Rademacher Complexity:

$$
\begin{aligned}
\mathbb{E}(\text{I}) &\leq 2 \mathbb{E}_\sigma \mathbb{E}_{\hat{\mathcal{S}}_1^T} \sup_h \sum_{t=1}^{T} \boldsymbol{\lambda}[t] \sum_{(x_t, y_t) \in \hat{\mathcal{S}}_t} \frac{1}{TN} \left( \alpha_t(y) \ell(h(x_t, y_t)) \right) \\
&\leq 2 \sum_t \boldsymbol{\lambda}[t] \mathbb{E}_\sigma \mathbb{E}_{\hat{\mathcal{S}}_1^T} \sup_h \sum_{(x_t, y_t) \in \hat{\mathcal{S}}_t} \frac{1}{TN} \left( \alpha_t(y) \ell(h(x_t, y_t)) \right) \\
&\leq 2 \sup_t \mathbb{E}_\sigma \mathbb{E}_{\hat{\mathcal{S}}_t} \sup_h \sum_{(x_t, y_t) \in \hat{\mathcal{S}}_t} \frac{1}{TN} \left[ \alpha_t(y) \ell(h(x_t, y_t)) \right] \\
&= \sup_t 2 \mathcal{R}_t(\ell, \mathcal{H}) = 2 \bar{R}(\ell, \mathcal{H})
\end{aligned}
$$

Where $\bar{R}(\ell, \mathcal{H}) = \sup_t \mathcal{R}_t(\ell, \mathcal{H}) = \sup_t \sup_{h \sim \mathcal{H}} \mathbb{E}_{\hat{\mathcal{S}}_t, \sigma} \sum_{(x_t, y_t) \in \hat{\mathcal{S}}_t} \frac{1}{TN} \left[ \alpha_t(y) \ell(h(x_t, y_t)) \right]$, represents the Rademacher complexity w.r.t. the prediction loss $\ell$, hypothesis $h$ and *true* label distribution ratio $\alpha_t$.

Therefore with high probability $1 - \delta$, we have:

$$\sup_h \left| \sum_t \boldsymbol{\lambda}[t] R_{\mathcal{S}}^{\alpha_t}(h) - \sum_t \boldsymbol{\lambda}[t] \hat{R}_{\mathcal{S}}^{\alpha_t}(h) \right| \leq \bar{\mathcal{R}}(\ell, h) + L_{\max} d_\infty^{\sup} \sqrt{\sum_{t=1}^{T} \frac{\boldsymbol{\lambda}[t]^2}{\beta_t}} \sqrt{\frac{\log(1/\delta)}{2N}}$$

**Bounding Term** $(\text{II})$    For all the hypothesis $h$, we have:

$$
\begin{aligned}
\left| \sum_t \boldsymbol{\lambda}[t] \hat{R}_{\mathcal{S}_t}^{\alpha_t}(h) - \sum_t \boldsymbol{\lambda}[t] \hat{R}_{\mathcal{S}_t}^{\hat{\alpha}_t}(h) \right| &= \left| \sum_t \boldsymbol{\lambda}[t] \frac{1}{N_{\mathcal{S}_t}} \sum_i^{N_{\mathcal{S}_t}} (\alpha(y(i)) - \hat{\alpha}(y(i))) \ell(h) \right| \\
&= \sum_t \boldsymbol{\lambda}[t] \frac{1}{N_{\mathcal{S}_t}} \left| \sum_y^{|\mathcal{Y}|} (\alpha(Y = y) - \hat{\alpha}(Y = y)) \bar{\ell}(Y = y) \right|
\end{aligned}
$$

Where $\bar{\ell}(Y = y) = \sum_i^{N_{\mathcal{S}_t}} \ell(h(x_i, y_i = y))$, represents the cumulative error, conditioned on a given label $Y = y$. According to the Holder inequality, we have:

$$
\begin{aligned}
\sum_t \boldsymbol{\lambda}[t] \frac{1}{N_{\mathcal{S}_t}} \left| \sum_y^{|\mathcal{Y}|} (\alpha_t(Y = y) - \hat{\alpha}_t(Y = y)) \bar{\ell}(Y = y) \right| &\leq \sum_t \boldsymbol{\lambda}[t] \frac{1}{N_{\mathcal{S}_t}} \|\alpha_t - \hat{\alpha}_t\|_2 \|\bar{\ell}(Y = y)\|_2 \\
&\leq L_{\max} \sum_t \boldsymbol{\lambda}[t] \|\alpha_t - \hat{\alpha}_t\|_2 \\
&\leq L_{\max} \sup_t \|\alpha_t - \hat{\alpha}_t\|_2
\end{aligned}
$$

Therefore, $\forall h \in \mathcal{H}$, with high probability $1 - \delta$ we have:

$$\sum_t \boldsymbol{\lambda}[t] R_{\mathcal{S}}^{\alpha_t}(h) \leq \sum_t \boldsymbol{\lambda}[t] \hat{R}_{\mathcal{S}}^{\hat{\alpha}_t}(h) + 2 \bar{\mathcal{R}}(\ell, h) + L_{\max} d_\infty^{\sup} \sqrt{\sum_{t=1}^{T} \frac{\boldsymbol{\lambda}[t]^2}{\beta_t}} \sqrt{\frac{\log(1/\delta)}{2N}} + L_{\max} \sup_t \|\alpha_t - \hat{\alpha}_t\|_2$$

### E.2 BOUNDING EMPIRICAL WASSERSTEIN DISTANCE

Then we need to derive the sample complexity of the empirical and true distributions, which can be decomposed as the following two parts. For any $t$, we have:

$$\mathbb{E}_{y\sim\mathcal{T}(y)}W_1(\mathcal{T}(x|Y=y)\|\mathcal{S}_t(x|Y=y)) - \mathbb{E}_{y\sim\hat{\mathcal{T}}(y)}W_1(\hat{\mathcal{T}}(x|Y=y)\|\hat{\mathcal{S}}_t(x|Y=y))$$

$$\leq \underbrace{\mathbb{E}_{y\sim\mathcal{T}(y)}W_1(\mathcal{T}(x|Y=y)\|\mathcal{S}_t(x|Y=y)) - \mathbb{E}_{y\sim\mathcal{T}(y)}W_1(\hat{\mathcal{T}}(x|Y=y)\|\hat{\mathcal{S}}_t(x|Y=y))}_{\text{(I)}}$$

$$+ \underbrace{\mathbb{E}_{y\sim\mathcal{T}(y)}W_1(\hat{\mathcal{T}}(x|Y=y)\|\hat{\mathcal{S}}_t(x|Y=y)) - \mathbb{E}_{y\sim\hat{\mathcal{T}}(y)}W_1(\hat{\mathcal{T}}(x|Y=y)\|\hat{\mathcal{S}}_t(x|Y=y))}_{\text{(II)}}$$

**Bounding (I)** We have:

$$\mathbb{E}_{y\sim\mathcal{T}(y)}W_1(\mathcal{T}(x|Y=y)\|\mathcal{S}_t(x|Y=y)) - \mathbb{E}_{y\sim\mathcal{T}(y)}W_1(\hat{\mathcal{T}}(x|Y=y)\|\hat{\mathcal{S}}_t(x|Y=y))$$

$$= \sum_y \mathcal{T}(y)\left(W_1(\mathcal{T}(x|Y=y)\|\mathcal{S}_t(x|Y=y)) - W_1(\hat{\mathcal{T}}(x|Y=y)\|\hat{\mathcal{S}}_t(x|Y=y))\right)$$

$$\leq |\sum_y \mathcal{T}(y)|\sup_y\left(W_1(\mathcal{T}(x|Y=y)\|\mathcal{S}_t(x|Y=y)) - W_1(\hat{\mathcal{T}}(x|Y=y)\|\hat{\mathcal{S}}_t(x|Y=y))\right)$$

$$= \sup_y\left(W_1(\mathcal{T}(x|Y=y)\|\mathcal{S}_t(x|Y=y)) - W_1(\hat{\mathcal{T}}(x|Y=y)\|\hat{\mathcal{S}}_t(x|Y=y))\right)$$

$$\leq \sup_y\ [W_1(\mathcal{S}_t(x|Y=y)\|\hat{\mathcal{S}}_t(x|Y=y)) + W_1(\hat{\mathcal{S}}_t(x|Y=y)\|\hat{\mathcal{T}}(x|Y=y))$$

$$+ W_1(\hat{\mathcal{T}}(x|Y=y)\|\mathcal{T}(x|Y=y)) - W_1(\hat{\mathcal{T}}(x|Y=y)\|\hat{\mathcal{S}}_t(x|Y=y))]$$

$$= \sup_y W_1(\mathcal{S}_t(x|Y=y)\|\hat{\mathcal{S}}_t(x|Y=y)) + W_1(\hat{\mathcal{T}}(x|Y=y)\|\mathcal{T}(x|Y=y))$$

The first inequality holds because of the Holder inequality. As for the second inequality, we use the triangle inequality of Wasserstein distance. $W_1(P\|Q) \leq W_1(P\|P_1) + W_1(P_1\|P_2) + W_1(P_2\|Q)$.

According to the convergence behavior of Wasserstein distance (Weed et al., 2019), with high probability $\geq 1 - 2\delta$ we have:

$$W_1(\mathcal{S}_t(x|Y=y)\|\hat{\mathcal{S}}_t(x|Y=y)) + W_1(\hat{\mathcal{T}}(x|Y=y)\|\mathcal{T}(x|Y=y)) \leq \kappa(\delta, N^y_{\mathcal{S}_t}, N^y_{\mathcal{T}})$$

Where $k(\delta, N^y_{\mathcal{S}_t}, N^y_{\mathcal{T}}) = C_{t,y}(N^y_{\mathcal{S}_t})^{-s_{t,y}} + C_y(N^y_{\mathcal{T}})^{-s_y} + \sqrt{\frac{1}{2}\log(\frac{2}{\delta})}(\sqrt{\frac{1}{N^y_{\mathcal{S}_t}}} + \sqrt{\frac{1}{N^y_t}})$, where $N^y_{\mathcal{S}_t}$ is the number of $Y=y$ in source $t$ and $N^y_{\mathcal{T}}$ is the number of $Y=y$ in target distribution. $C_{t,y}$, $C_y$ $s_{t,y} > 2$, $s_y > 2$ are positive constant in the concentration inequality. This indicates the convergence behavior between empirical and true Wasserstein distance.

If we adopt the union bound (over all the labels) by setting $\delta \leftarrow \delta/|\mathcal{Y}|$, then with high probability $\geq 1 - 2\delta$, we have:

$$\sup_y W_1(\mathcal{S}(x|Y=y)\|\hat{\mathcal{S}}(x|Y=y)) + W_1(\hat{\mathcal{T}}(x|Y=y)\|\mathcal{T}(x|Y=y)) \leq \kappa(\delta, N^y_{\mathcal{S}_t}, N^y_{\mathcal{T}})$$

where $\kappa(\delta, N^y_{\mathcal{S}_t}, N^y_{\mathcal{T}}) = C_{t,y}(N^y_{\mathcal{S}_t})^{-s_{t,y}} + C_y(N^y_{\mathcal{T}})^{-s_y} + \sqrt{\frac{1}{2}\log(\frac{2|\mathcal{Y}|}{\delta})}(\sqrt{\frac{1}{N^y_{\mathcal{S}_t}}} + \sqrt{\frac{1}{N^y_{\mathcal{T}}}})$

Again by adopting the union bound (over all the tasks) by setting $\delta \leftarrow \delta/T$, with high probability $\geq 1 - 2\delta$, we have:

$$\sum_t \boldsymbol{\lambda}[t]\mathbb{E}_{y\sim\mathcal{T}(y)}W_1(\mathcal{T}(x|Y=y)\|\mathcal{S}(x|Y=y)) - \sum_t \boldsymbol{\lambda}[t]\mathbb{E}_{y\sim\mathcal{T}(y)}W_1(\hat{\mathcal{T}}(x|Y=y)\|\hat{\mathcal{S}}(x|Y=y)) \leq \sup_t \kappa(\delta, N^y_{\mathcal{S}_t}, N^y_{\mathcal{T}})$$

Where $\kappa(\delta, N^y_{\mathcal{S}_t}, N^y_{\mathcal{T}}) = C_{t,y}(N^y_{\mathcal{S}_t})^{-s_{t,y}} + C_y(N^y_{\mathcal{T}})^{-s_y} + \sqrt{\frac{1}{2}\log(\frac{2T|\mathcal{Y}|}{\delta})}(\sqrt{\frac{1}{N^y_{\mathcal{S}_t}}} + \sqrt{\frac{1}{N^y_{\mathcal{T}}}})$.

**Bounding** (**II**)    We can bound the second term:

$$\mathbb{E}_{y \sim \mathcal{T}(y)} W_1(\hat{\mathcal{T}}(x|Y=y) \| \hat{\mathcal{S}}_t(x|Y=y)) - \mathbb{E}_{y \sim \hat{\mathcal{T}}(y)} W_1(\hat{\mathcal{T}}(x|Y=y) \| \hat{\mathcal{S}}_t(x|Y=y))$$

$$\leq \sup_y W_1(\hat{\mathcal{T}}(x|Y=y) \| \hat{\mathcal{S}}_t(x|Y=y)) | \sum_y \mathcal{T}(y) - \hat{\mathcal{T}}(y)|$$

$$\leq C_{\max}^t | \sum_y \mathcal{T}(y) - \hat{\mathcal{T}}(y)|$$

Where $C_{\max}^t = \sup_y W_1(\hat{\mathcal{T}}(x|Y=y) \| \hat{\mathcal{S}}(x|Y=y))$ is a positive and bounded constant. Then we need to bound $|\sum_y \mathcal{T}(y) - \hat{\mathcal{T}}(y)|$, by adopting MicDiarmid's inequality, we have at high probability $1 - \delta$:

$$|\sum_y \mathcal{T}(y) - \hat{\mathcal{T}}(y)| \leq \mathbb{E}_{\hat{\mathcal{T}}} | \sum_y \mathcal{T}(y) - \hat{\mathcal{T}}(y)| + \sqrt{\frac{\log(1/\delta)}{2N_{\mathcal{T}}}}$$

$$= 2\mathbb{E}_\sigma \mathbb{E}_{\hat{\mathcal{T}}} \sum_y \sigma \hat{\mathcal{T}}(y) + \sqrt{\frac{\log(1/\delta)}{2N_{\mathcal{T}}}}$$

Then we bound $\mathbb{E}_\sigma \mathbb{E}_{\hat{\mathcal{T}}} \sum_y \sigma \hat{\mathcal{T}}(y)$. We use the properties of Rademacher complexity [Lemma 26.11, (Shalev-Shwartz & Ben-David, 2014)] and notice that $\hat{\mathcal{T}}(y)$ is a probability simplex, then we have:

$$\mathbb{E}_\sigma \mathbb{E}_{\hat{\mathcal{T}}} \sum_y \sigma \hat{\mathcal{T}}(y) \leq \sqrt{\frac{2\log(2|\mathcal{Y}|)}{N_{\mathcal{T}}}}$$

Then we have $|\sum_y \mathcal{T}(y) - \hat{\mathcal{T}}(y)| \leq \sqrt{\frac{2\log(2|\mathcal{Y}|)}{N_{\mathcal{T}}}} + \sqrt{\frac{\log(1/\delta)}{2N_{\mathcal{T}}}}$

Then using the union bound and denoting $\delta \leftarrow \delta/T$, with high probability $\geq 1 - \delta$ and for any simplex $\boldsymbol{\lambda}$, we have:

$$\sum_t \boldsymbol{\lambda}[t] \mathbb{E}_{y \sim \mathcal{T}(y)} W_1(\hat{\mathcal{T}}(x|Y=y) \| \hat{\mathcal{S}}_t(x|Y=y)) \leq \sum_t \boldsymbol{\lambda}[t] \mathbb{E}_{y \sim \hat{\mathcal{T}}(y)} W_1(\hat{\mathcal{T}}(x|Y=y) \| \hat{\mathcal{S}}_t(x|Y=y))$$

$$C_{\max}(\sqrt{\frac{2\log(2|\mathcal{Y}|)}{N_{\mathcal{T}}}} + \sqrt{\frac{\log(T/\delta)}{2N_{\mathcal{T}}}})$$

where $C_{\max} = \sup_t C_{\max}^t$.

Combining together, we can derive the PAC-Learning bound, which is estimated from the finite samples (with high probability $1 - 4\delta$):

$$R_{\mathcal{T}}(h) \leq \sum_t \boldsymbol{\lambda}_t \hat{R}_{\mathcal{S}_t}^{\hat{\alpha}_t}(h) + LH \sum_t \boldsymbol{\lambda}_t \mathbb{E}_{y \sim \hat{\mathcal{T}}(y)} W_1(\hat{\mathcal{T}}(x|Y=y) \| \hat{\mathcal{S}}(x|Y=y)) + L_{\max} d_\infty^{\sup} \sqrt{\sum_{t=1}^T \frac{\boldsymbol{\lambda}_t^2}{\beta_t}} \sqrt{\frac{\log(1/\delta)}{2N}}$$

$$+ 2\bar{\mathcal{R}}(\ell, h) + L_{\max} \sup_t \|\alpha_t - \hat{\alpha}_t\|_2 + \sup_t \kappa(\delta, N_{\mathcal{S}_t}^y, N_{\mathcal{T}}^y) + C_{\max}(\sqrt{\frac{2\log(2|\mathcal{Y}|)}{N_{\mathcal{T}}}} + \sqrt{\frac{\log(T/\delta)}{2N_{\mathcal{T}}}})$$

Then we denote $\text{Comp}(N_{\mathcal{S}_1}, \ldots, N_{\mathcal{T}}, \delta) = 2\bar{\mathcal{R}}(\ell, h) + \sup_t \kappa(\delta, N_{\mathcal{S}_t}^y, N_{\mathcal{T}}^y) + C_{\max}(\sqrt{\frac{2\log(2|\mathcal{Y}|)}{N_{\mathcal{T}}}} + \sqrt{\frac{\log(T/\delta)}{2N_{\mathcal{T}}}})$ as the convergence rate function that decreases with larger $N_{\mathcal{S}_1}, \ldots, N_{\mathcal{T}}$. Bedsides, $\bar{\mathcal{R}}(\ell, h) = \sup_t \mathcal{R}_t(\ell, \mathcal{H})$ is the re-weighted Rademacher complexity. Given a fixed hypothesis with finite VC dimension [2], it can be proved $\bar{\mathcal{R}}(\ell, h) = \min_{N_{\mathcal{S}_1}, \ldots, N_{\mathcal{S}_T}} \mathcal{O}(\sqrt{\frac{1}{N_{\mathcal{S}_t}}})$ i.e (Shalev-Shwartz & Ben-David, 2014). $\qquad \square$

---

[2] If the hypothesis is the neural network, the Rademacher complexity can still be bounded analogously.

# F    PROOF OF THEOREM 2

We first recall the stochastic feature representation $g$ such that $g : \mathcal{X} \to \mathcal{Z}$ and *scoring hypothesis* h $h : \mathcal{Z} \times \mathcal{Y} \to \mathbb{R}$ and the prediction loss $\ell$ with $\ell : \mathbb{R} \to \mathbb{R}$. [3]

*Proof.* The marginal distribution and conditional distribution w.r.t. latent variable $Z$ that are induced by $g$, which can be reformulated as:

$$\mathcal{S}(z) = \int_x g(z|x)\mathcal{S}(x)dx \qquad \mathcal{S}(z|y) = \int_x g(z|x)\mathcal{S}(x|Y=y)dx$$

In the multi-class classification problem, we additionally define the following distributions:

$$\mu^k(z) = \mathcal{S}(Y=k, z) = \mathcal{S}(Y=k)\mathcal{S}(z|Y=k)$$
$$\pi^k(z) = \mathcal{T}(Y=k, z) = \mathcal{T}(Y=k)\mathcal{T}(z|Y=k)$$

Based on (Nguyen et al., 2009) and $g(z|x)$ is a stochastic representation learning function, the loss conditioned a fixed point $(x, y)$ w.r.t. $h$ and $g$ is $\mathbb{E}_{z \sim g(z|x)}\ell(h(z, y))$. Then taking the expectation over the $\mathcal{S}(x, y)$ we have: [4]

$$
\begin{aligned}
R_{\mathcal{S}}(h, g) &= \mathbb{E}_{(x,y) \sim \mathcal{S}(x,y)} \mathbb{E}_{z \sim g(z|x)} \ell(h(z, y)) \\
&= \sum_{k=1}^{|\mathcal{Y}|} \mathcal{S}(y=k) \int_x \mathcal{S}(x|Y=k) \int_z g(z|x)\ell(h(z, y=k))dzdx \\
&= \sum_{k=1}^{|\mathcal{Y}|} \mathcal{S}(y=k) \int_z [\int_x \mathcal{S}(x|Y=k)g(z|x)dx]\ell(h(z, y=k))dz \\
&= \sum_{k=1}^{|\mathcal{Y}|} \mathcal{S}(y=k) \int_z \mathcal{S}(z|Y=k)\ell(h(z, y=k))dz \\
&= \sum_{k=1}^{|\mathcal{Y}|} \int_z \mathcal{S}(z, Y=k)\ell(h(z, y=k))dz \\
&= \sum_{k=1}^{|\mathcal{Y}|} \int_z \mu^k(z)\ell(h(z, y=k))dz
\end{aligned}
$$

Intuitively, the expected loss w.r.t. the joint distribution $\mathcal{S}$ can be decomposed as the expected loss on the label distribution $\mathcal{S}(y)$ (weighted by the labels) and conditional distribution $\mathcal{S}(\cdot|y)$ (real valued conditional loss).

Then the expected risk on the $\mathcal{S}$ and $\mathcal{T}$ can be expressed as:

$$R_{\mathcal{S}}(h, g) = \sum_{k=1}^{|\mathcal{Y}|} \int_z \ell(h(z, y=k))\mu^k(z)dz$$

$$R_{\mathcal{T}}(h, g) = \sum_{k=1}^{|\mathcal{Y}|} \int_z \ell(h(z, y=k))\pi^k(z)dz$$

---

[3]Note this definition is different from the conventional binary classification with binary output, and it is more suitable in the multi-classification scenario and cross entropy loss (Hoffman et al., 2018a). For example, if we define $l = -\log(\cdot)$ and $h(z, y) \in (0, 1)$ as a scalar score output. Then $\ell(h(z, y))$ can be viewed as the cross-entropy loss for the neural-network.

[4]An alternative understanding is based on the Markov chain. In this case it is a DAG with $Y \xleftarrow{\mathcal{S}(y|x)} X \xrightarrow{g} Z$, $X \xrightarrow{\mathcal{S}(y|x)} Y \xrightarrow{h} S \xleftarrow{h} Z \xleftarrow{g} X$. (S is the output of the scoring function). Then the expected loss over the all random variable can be equivalently written as $\int \mathbb{P}(x, y, z, s) \ell(s) d(x, y, z, s) = \int \mathbb{P}(x)\mathbb{P}(y|x)\mathbb{P}(z|x)\mathbb{P}(s|z, y)\ell(s) = \int \mathbb{P}(x, y)\mathbb{P}(z|x)\mathbb{P}(s|z, y)\ell(s)d(x, y)d(z)d(s)$. Since the scoring $S$ is determined by $h(x, y)$, then $\mathbb{P}(s|y, z) = 1$. According to the definition we have $\mathbb{P}(z|x) = g(z|x)$, $\mathbb{P}(x, y) = \mathcal{S}(x, y)$, then the loss can be finally expressed as $\mathbb{E}_{\mathcal{S}(x,y)} \mathbb{E}_{g(z|x)} \ell(h(z, y))$

By denoting $\alpha(y) = \frac{\mathcal{T}(y)}{\mathcal{S}(y)}$, we have the $\alpha$-weighted loss:

$$R^\alpha_\mathcal{S}(h,g) = \mathcal{T}(Y=1)\int_z \ell(h(z,y=1))\mathcal{S}(z|Y=1) + \mathcal{T}(Y=2)\int_z \ell(h(z,y=2))\mathcal{S}(z|Y=2)$$
$$+ \cdots + \mathcal{T}(Y=k)\int_z \ell(h(z,y=k))\mathcal{S}(z|Y=k)dz$$

Then we have:

$$R_\mathcal{T}(h,g) - R^\alpha_\mathcal{S}(h,g) \le \sum_k \mathcal{T}(Y=k)\int_z \ell(h(z,y=k))d|\mathcal{S}(z|Y=k) - \mathcal{T}(z|Y=k)|$$

Under the same assumption, we have the loss function $\ell(h(z, Y=k))$ is KL-Lipschitz w.r.t. the cost $\|\cdot\|_2$ (given a fixed $k$). Therefore by adopting the same proof strategy (Kantorovich-Rubinstein duality) in Lemma 2, we have

$$\le KL\mathcal{T}(Y=1)W_1(\mathcal{S}(z|Y=1)\|\mathcal{T}(z|Y=1)) + \cdots + KL\mathcal{T}(Y=k)W_1(\mathcal{S}(z|Y=k)\|\mathcal{T}(z|Y=k))$$
$$= KL\mathbb{E}_{y\sim\mathcal{T}(y)}W_1(\mathcal{S}(z|Y=y)\|\mathcal{T}(z|Y=y))$$

Therefore, we have:

$$R_\mathcal{T}(h,g) \le R^\alpha_\mathcal{S}(h,g) + LK\mathbb{E}_{y\sim\mathcal{T}(y)}W_1(\mathcal{S}(z|Y=y)\|\mathcal{T}(z|Y=y))$$

Based on the aforementioned result, we have $\forall t = 1, \ldots, T$ and denote $\mathcal{S} = \mathcal{S}_t$ and $\alpha(y) = \alpha_t(y) = \mathcal{T}(y)/\mathcal{S}_t(y)$:

$$\boldsymbol{\lambda}[t]R_\mathcal{T}(h,g) \le \boldsymbol{\lambda}[t]R^{\alpha_t}_{\mathcal{S}_t}(h,g) + LK\boldsymbol{\lambda}[t]\mathbb{E}_{y\sim\mathcal{T}(y)}W_1(\mathcal{S}_t(z|Y=y)\|\mathcal{T}(z|Y=y))$$

Summing over $t = 1, \ldots, T$, we have:

$$R_\mathcal{T}(h,g) \le \sum_{t=1}^T \boldsymbol{\lambda}[t]R^{\alpha_t}_{\mathcal{S}_t}(h,g) + LK\sum_{t=1}^T \boldsymbol{\lambda}[t]\mathbb{E}_{y\sim\mathcal{T}(y)}W_1(\mathcal{S}_t(z|Y=y)\|\mathcal{T}(z|Y=y))$$

$\square$

## G  APPROXIMATION $W_1$ DISTANCE

According to Jensen inequality, we have

$$W_1(\hat{\mathcal{S}}_t(z|Y=y)\|\hat{\mathcal{T}}(z|Y=y)) \le \sqrt{[W_2(\hat{\mathcal{S}}_t(z|Y=y)\|\hat{\mathcal{T}}(z|Y=y))]^2}$$

Supposing $\hat{\mathcal{S}}_t(z|Y=y) \approx \mathcal{N}(\mathbf{C}^y_t, \boldsymbol{\Sigma})$ and $\hat{\mathcal{T}}(z|Y=y) \approx \mathcal{N}(\mathbf{C}^y, \boldsymbol{\Sigma})$, then we have:

$$[W_2(\hat{\mathcal{S}}_t(z|Y=y)\|\hat{\mathcal{T}}(z|Y=y)]^2 = \|\mathbf{C}^y_t - \mathbf{C}^y\|^2_2 + \text{Trace}(2\boldsymbol{\Sigma} - 2(\boldsymbol{\Sigma}\boldsymbol{\Sigma})^{1/2}) = \|\mathbf{C}^y_t - \mathbf{C}^y\|^2_2$$

We would like to point out that assuming the identical covariance matrix is more computationally efficient during the matching. This is advantageous and reasonable in the deep learning regime: we adopted the mini-batch (ranging from 20-128) for the neural network parameter optimization, in each mini-batch the samples of each class are **small**, then we compute the empirical covariance/variance matrix will be surely **biased** to the ground truth variance and induce a much higher complexity to optimize. By the contrary, the empirical mean is **unbiased** and computationally efficient, we can simply use the moving the moving average to efficiently update the estimated mean value (with a unbiased estimator). The empirical results verify the effectiveness of this idea.

## H  PROOF OF LEMMA 1

For each source $\mathcal{S}_t$, by introducing the duality of Wasserstein-1 distance, for $y \in \mathcal{Y}$, we have:

$$W_1(\mathcal{S}_t(z|y)\|\mathcal{T}(z|y)) = \sup_{\|d\|_L \le 1} \mathbb{E}_{z\sim\mathcal{S}_t(z|y)}d(z) - \mathbb{E}_{z\sim\mathcal{T}(z|y)}d(z)$$
$$= \sup_{\|d\|_L \le 1} \sum_z \mathcal{S}_t(z|y)d(z) - \sum_z \mathcal{T}(z|y)d(z)$$
$$= \frac{1}{\mathcal{T}(y)}\sup_{\|d\|_L \le 1} \frac{\mathcal{T}(y)}{\mathcal{S}_t(y)}\sum_z \mathcal{S}_t(z,y)d(z) - \sum_z \mathcal{T}(z,y)d(z)$$

Then by defining $\bar{\alpha}_t(z) = \mathbf{1}_{\{(z,y)\sim\mathcal{S}_t\}}\frac{\mathcal{T}(Y=y)}{\mathcal{S}_t(Y=y)} = \mathbf{1}_{\{(z,y)\sim\mathcal{S}_t\}}\alpha_t(Y=y)$, we can see for each pair observation $(z,y)$ sampled from the same distribution, then $\bar{\alpha}_t(Z=z) = \alpha_t(Y=y)$. Then we have:

$$\sum_y \mathcal{T}(y)W_1(\mathcal{S}_t(z|y)\|\mathcal{T}(z|y)) = \sum_y \sup_{\|d\|_L\leq 1}\{\sum_z \alpha_t(y)\mathcal{S}_t(z,y)d(z) - \sum_z \mathcal{T}(z,y)d(z)\}$$

$$= \sup_{\|d\|_L\leq 1}\sum_z \bar{\alpha}_t(z)\mathcal{S}_t(z)d(z) - \sum_z \mathcal{T}(z)d(z)$$

$$= \sup_{\|d\|_L\leq 1}\mathbb{E}_{z\sim\mathcal{S}_t(z)}\bar{\alpha}_t(z)d(z) - \mathbb{E}_{z\sim\mathcal{T}(z)}d(z)$$

We propose a simple example to understand $\bar{\alpha}_t$: supposing three samples in $\mathcal{S}_t = \{(z_1, Y = 1), (z_2, Y = 1), (z_3, Y = 0)\}$ then $\bar{\alpha}_t(z_1) = \bar{\alpha}_t(z_2) = \alpha_t(1)$ and $\bar{\alpha}_t(z_3) = \alpha_t(0)$. Therefore, the conditional term is equivalent to the label-weighted Wasserstein adversarial learning. We plug in each source domain as weight $\boldsymbol{\lambda}[t]$ and domain discriminator as $d_t$, we finally have Lemma 1.

## I    DERIVE THE LABEL RATIO LOSS

We suppose the representation learning aims at matching the conditional distribution such that $\mathcal{T}(z|y) \approx \mathcal{S}_t(z|y), \forall t$, then we suppose the predicted target distribution as $\bar{\mathcal{T}}(y)$. By simplifying the notation, we define $f(z) = \arg\max_y h(z,y)$ the most possible prediction label output, then we have:

$$\bar{\mathcal{T}}(y) = \sum_{k=1}^{\mathcal{Y}} \mathcal{T}(f(z) = y|Y = k)\mathcal{T}(Y = k) = \sum_{k=1}^{\mathcal{Y}} \mathcal{S}_t(f(z) = y|Y = k)\mathcal{T}(Y = k)$$

$$= \sum_{i=1}^{\mathcal{Y}} \mathcal{S}_t(f(z) = y, Y = k)\alpha_t(k) = \bar{\mathcal{T}}_{\alpha_t}(y)$$

The first equality comes from the definition of target label prediction distribution, $\bar{\mathcal{T}}(y) = \mathbb{E}_{\mathcal{T}(z)}\mathbf{1}\{f(z) = y\} = \mathcal{T}(f(z) = y) = \sum_{k=1}^{\mathcal{Y}} \mathcal{T}(f(z) = y, Y = k) = \sum_{k=1}^{\mathcal{Y}} \mathcal{T}(f(z) = y|Y = k)\mathcal{T}(Y = k)$.

The second equality $\mathcal{T}(f(z) = y|Y = k) = \mathcal{S}_t(f(z) = y|Y = k)$ holds since $\forall t$, $\mathcal{T}(z|y) \approx \mathcal{S}_t(z|y)$, then for the shared hypothesis $f$, we have $\mathcal{T}(f(z) = y|Y = k) = \mathcal{S}_t(f(z) = y|Y = k)$.

The term $\mathcal{S}_t(f(z) = y, Y = k)$ is the (expected) source prediction confusion matrix, and we denote its empirical (observed) version as $\hat{\mathcal{S}}_t(f(z) = y, Y = k)$.

Based on this idea, in practice we want to find a $\hat{\alpha}_t$ to match the two predicted distribution $\bar{\mathcal{T}}$ and $\bar{\mathcal{T}}_{\hat{\alpha}_t}$. If we adopt the KL-divergence as the metric, we have:

$$\min_{\hat{\alpha}_t} D_{\text{KL}}(\bar{\mathcal{T}}\|\bar{\mathcal{T}}_{\hat{\alpha}_t}) = \min_{\hat{\alpha}_t} \mathbb{E}_{y\sim\bar{\mathcal{T}}}\log(\frac{\bar{\mathcal{T}}(y)}{\bar{\mathcal{T}}_{\hat{\alpha}_t}(y)}) = \min_{\hat{\alpha}_t} -\mathbb{E}_{y\sim\bar{\mathcal{T}}}\log(\bar{\mathcal{T}}_{\hat{\alpha}_t}(y))$$

$$= \min_{\hat{\alpha}_t} -\sum_y \bar{\mathcal{T}}(y)\log(\sum_{k=1}^{\mathcal{Y}} \mathcal{S}_t(f(z) = y, Y = k)\hat{\alpha}_t(k))$$

We should notice the nature constraints of label ratio: $\{\hat{\alpha}_t(y) \geq 0, \sum_y \hat{\alpha}_t(y)\hat{\mathcal{S}}_t(y) = 1\}$. Based on this principle, we proposed the optimization problem to estimate each label ratio. We adopt its empirical counterpart, the empirical confusion matrix $C_{\hat{\mathcal{S}}_t}[y,k] = \hat{\mathcal{S}}_t[f(z) = y, Y = k]$, then the optimization loss can be expressed as:

$$\min_{\hat{\alpha}_t} \quad -\sum_{y=1}^{|\mathcal{Y}|} \bar{\mathcal{T}}(y)\log(\sum_{k=1}^{|\mathcal{Y}|} C_{\hat{\mathcal{S}}_t}[y,k]\hat{\alpha}_t(k))$$

$$\text{s.t.} \quad \forall y \in \mathcal{Y}, \hat{\alpha}_t(y) \geq 0, \quad \sum_y \hat{\alpha}_t(y)\hat{\mathcal{S}}_t(y) = 1$$

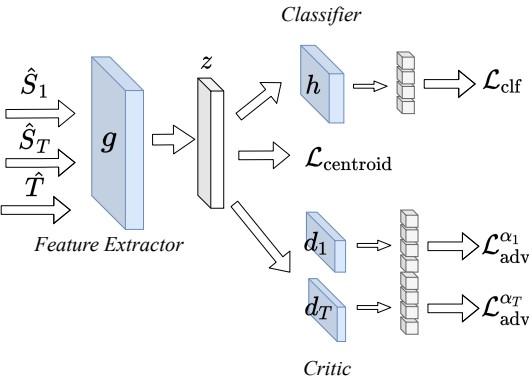

Figure 4: Network Structure of Proposed Approach. It consists three losses: the weighted Classification losses; the centroid matching for explicit conditional matching; the weighted adversarial loss for implicit conditional matching, showed in Eq. (6)

## J  LABEL PARTIAL MULTI-SOURCE UNSUPERVISED DA

The key difference between multi-conventional and partial unsupervised DA is the estimation step of $\hat{\alpha}_t$. In fact, we only add a sparse constraint for estimating each $\hat{\alpha}_t$:

$$
\min_{\hat{\alpha}_t} \quad -\sum_{y=1}^{|\mathcal{Y}|} \bar{\mathcal{T}}(y) \log(\sum_{k=1}^{|\mathcal{Y}|} C_{\hat{\mathcal{S}}_t}[y,k]\hat{\alpha}_t(k)) + C_2\|\hat{\alpha}_t\|_1
$$

$$
\text{s.t.} \quad \forall y \in \mathcal{Y}, \hat{\alpha}_t(y) \geq 0, \quad \sum_y \hat{\alpha}_t(y)\hat{\mathcal{S}}_t(y) = 1
$$

(5)

Where $C_2$ is the hyper-parameter to control the level of target label sparsity, to estimate the target label distribution. In the paper, we denote $C_2 = 0.1$.

## K  EXPLICIT AND IMPLICIT CONDITIONAL LEARNING

Inspired by Theorem 2, we need to learn the function $g : \mathcal{X} \to \mathcal{Z}$ and $h : \mathcal{Z} \times \mathcal{Y} \to \mathbb{R}$ to minimize:

$$
\min_{g,h} \sum_t \boldsymbol{\lambda}[t]\hat{R}_{\mathcal{S}_t}^{\hat{\alpha}_t}(h,g) + C_0 \sum_t \boldsymbol{\lambda}[t]\mathbb{E}_{y\sim\hat{\mathcal{T}}(y)} W_1(\hat{\mathcal{S}}_t(z|Y=y)\|\hat{\mathcal{T}}(z|Y=y))
$$

This can be equivalently expressed as:

$$
\min_{g,h} \sum_t \boldsymbol{\lambda}[t]\hat{R}_{\mathcal{S}_t}^{\alpha_t}(h,g) + \epsilon C_0 \sum_t \boldsymbol{\lambda}[t]\mathbb{E}_{y\sim\hat{\mathcal{T}}(y)} W_1(\hat{\mathcal{S}}_t(z|Y=y)\|\hat{\mathcal{T}}(z|Y=y))
$$
$$
+ (1-\epsilon)C_0 \sum_t \boldsymbol{\lambda}[t]\mathbb{E}_{y\sim\hat{\mathcal{T}}(y)} W_1(\hat{\mathcal{S}}_t(z|Y=y)\|\hat{\mathcal{T}}(z|Y=y))
$$

Due to the explicit and implicit approximation of conditional distance, we then optimize an alternative form:

$$
\min_{g,h} \max_{d_1,\dots,d_T} \underbrace{\sum_t \boldsymbol{\lambda}[t]\hat{R}_{\mathcal{S}_t}^{\hat{\alpha}_t}(h,g)}_{\text{Classification Loss}} + \epsilon C_0 \underbrace{\sum_t \boldsymbol{\lambda}[t]\mathbb{E}_{y\sim\hat{\mathcal{T}}(y)}\|\mathbf{C}_t^y - \mathbf{C}^y\|_2}_{\text{Explicit Conditional Loss}}
$$
$$
+ (1-\epsilon)C_0 \underbrace{\sum_t \boldsymbol{\lambda}[t][\mathbb{E}_{z\sim\hat{\mathcal{S}}_t(z)}\bar{\alpha}^t(z)d(z) - \mathbb{E}_{z\sim\hat{\mathcal{T}}(z)}d(z)]}_{\text{Implicit Conditional Loss}}
$$

(6)

Where

- $\mathbf{C}_t^y = \sum_{(z_t, y_t) \sim \hat{\mathcal{S}}_t} \mathbf{1}_{\{y_t = y\}} z_t$ the centroid of label $Y = y$ in source $\mathcal{S}_t$.

- $\mathbf{C}^y = \sum_{(z_t, y_p) \sim \hat{\mathcal{T}}} \mathbf{1}_{\{y_p = y\}} z_t$ the centroid of pseudo-label $Y = y_p$ in target $\mathcal{S}_t$. (If it is the unsupervised DA scenarios).

- $\bar{\alpha}_t(z) = \mathbf{1}_{\{(z,y) \sim \mathcal{S}_t\}} \hat{\alpha}_t(Y = y)$, namely if each pair observation $(z, y)$ from the distribution, then $\bar{\alpha}_t(Z = z) = \hat{\alpha}_t(Y = y)$.

- $d_1, \cdots, d_T$ are domain discriminator (or critic function) restricted within 1-Lipschitz function.

- $\epsilon \in [0, 1]$ is the adjustment parameter in the trade-off of explicit and implicit learning. Based on the equivalence form, our approach proposed a theoretical principled way to tuning its weights. In the paper, we assume $\epsilon = 0.5$.

- $\hat{\mathcal{T}}(y)$ empirical target label distribution. (In the unsupervised DA scenarios, we approximate it by predicted target label distribution $\bar{\mathcal{T}}(y)$.)

**Gradient Penalty**   In order to enforce the Lipschitz property of the statistic critic function, we adopt the gradient penalty term (Gulrajani et al., 2017). More concretely, given two samples $z_s \sim \mathcal{S}_t(z)$ and $z_t \sim \mathcal{T}(z)$ we generate an interpolated sample $z_{\text{int}} = \xi z_s + (1 - \xi) z_t$ with $\xi \sim \text{Unif}[0, 1]$. Then we add a gradient penalty $\|\nabla d(z_{\text{int}})\|_2^2$ as a regularization term to control the Lipschitz property w.r.t. the discriminator $d_1, \cdots, d_T$.

## L   ALGORITHM DESCRIPTIONS

We propose a detailed pipeline of the proposed algorithm in the following, shown in Algorithm 2 and 3. As for updating $\boldsymbol{\lambda}$ and $\alpha_t$, we iteratively solve the convex optimization problem after each training epoch and updating them by using the moving average technique.

For solving the $\boldsymbol{\lambda}$ and $\alpha_t$, we notice that frequently updating these two parameters in the mini-batch level will lead to an instability result during the training. [5] As a consequence, we compute the accumulated confusion matrix, weighted prediction risk, and conditional Wasserstein distance for the whole training epoch and then solve the optimization problem. We use CVXPY to optimize the two standard convex losses. [6]

**Comparison with different time and memory complexity.**   We discuss the time and memory complexity of our approach.

Time complexity: In computing each batch we need to compute $T$ re-weighted loss, $T$ domain adversarial loss and $T$ explicit conditional loss. Then our computational complexity is still $(O)(T)$ during the mini-batch training, which is comparable with recent SOTA such as MDAN and DARN. In addition, after each training epoch we need to estimate $\alpha_t$ and $\boldsymbol{\lambda}$, which can have time complexity $\mathcal{O}(T|\mathcal{Y}|)$ with each epoch. (If we adopt SGD to solve these two convex problems). Therefore, the our proposed algorithm is time complexity $\mathcal{O}(T|\mathcal{Y}|)$. The extra $\mathcal{Y}$ term in time complexity is due to the approach of label shift in the designed algorithm.

Memory Complexity: Our proposed approach requires $\mathcal{O}(T)$ domain discriminator and $\mathcal{O}(T|\mathcal{Y}|)$ class-feature centroids. By the contrary, MDAN and DARN require $\mathcal{O}(T)$ domain discriminator and M3SDA and MDMN require $\mathcal{O}(T^2)$ domain discriminators. Since our class-feature centroids are defined in the latent space ($z$), then the memory complexity of the class-feature centroids can be much smaller than domain discriminators.

---

[5]In the label distribution shift scenarios, the mini-batch datasets are highly labeled imbalanced. If we evaluate $\alpha_t$ over the mini-batch, it can be computationally expensive and unstable.

[6]The optimization problem w.r.t. $\alpha_t$ and $\boldsymbol{\lambda}$ is not large scale, then using the standard convex solver is fast and accurate.

---

**Algorithm 2** Wasserstein Aggregation Domain Network (unsupervised scenarios, one iteration)

---

**Require:** Labeled source samples $\hat{\mathcal{S}}_1, \ldots, \hat{\mathcal{S}}_T$, Target samples $\hat{\mathcal{T}}$
**Ensure:** Label distribution ratio $\hat{\alpha}_t$ and task relation simplex $\boldsymbol{\lambda}$. Feature Learner $g$, Classifier $h$, Statistic critic function $d_1, \ldots, d_T$, class centroid for source $\mathbf{C}_t^y$ and target $\mathbf{C}^y$ ($\forall t = [1, T], y \in \mathcal{Y}$).
  1: $\triangleright \triangleright \triangleright$ DNN Parameter Training Stage (fixed $\alpha_t$ and $\boldsymbol{\lambda}$) $\triangleleft \triangleleft \triangleleft$
  2: **for** mini-batch of samples $(\mathbf{x}_{\mathcal{S}_1}, \mathbf{y}_{\mathcal{S}_1}) \sim \hat{\mathcal{S}}_1, \ldots, (\mathbf{x}_{\mathcal{S}_T}, \mathbf{y}_{\mathcal{S}_T}) \sim \hat{\mathcal{S}}_T, (\mathbf{x}_{\mathcal{T}}) \sim \hat{\mathcal{T}}$ **do**
  3:      Predict target pseudo-label $\bar{\mathbf{y}}_{\mathcal{T}} = \text{argmax}_y h(g(\mathbf{x}_{\mathcal{T}}), y)$
  4:      Compute source confusion matrix for each batch (un-normalized)
          $C_{\hat{\mathcal{S}}_t} = \#[\text{argmax}_{y'} h(z, y') = y, Y = k]$ $(t = 1, \ldots, T)$
  5:      Compute the *batched* class centroid for source $C_t^y$ and target $C^y$.
  6:      Moving Average for update source/target class centroid: (We set $\epsilon_1 = 0.7$)
  7:          Source class centroid update     $\mathbf{C}_t^y = \epsilon_1 \times \mathbf{C}_t^y + (1 - \epsilon_1) \times C_t^y$
  8:          Target class centroid update     $\mathbf{C}^y = \epsilon_1 \times \mathbf{C}^y + (1 - \epsilon_1) \times C^y$
  9:      Updating $g, h, d_1, \ldots, d_T$ (SGD and Gradient Reversal), based on Eq.(6)
10: **end for**
11: $\triangleright \triangleright \triangleright$ Estimation $\hat{\alpha}_t$ and $\boldsymbol{\lambda}$ $\triangleleft \triangleleft \triangleleft$
12: Compute the global(normalized) source confusion matrix
     $C_{\hat{\mathcal{S}}_t} = \hat{\mathcal{S}}_t[\text{argmax}_{y'} h(z, y') = y, Y = k]$ $(t = 1, \ldots, T)$
13: Solve $\alpha_t$ (denoted as $\{\alpha_t'\}_{t=1}^T$) by Equation (2) (Or Eq.(5)) in the partial scenario).
14: Update $\alpha_t$ by moving average: $\alpha_t = \epsilon_1 \times \alpha_t + (1 - \epsilon_1) \times \alpha_t'$
15: Compute the weighted loss and weighted centroid distance, then solve $\boldsymbol{\lambda}$ (denoted as $\boldsymbol{\lambda}'$) from Sec. 2.3.
16: Updating $\boldsymbol{\lambda}$ by moving average: $\boldsymbol{\lambda} = 0.8 \times \boldsymbol{\lambda} + 0.2 \times \boldsymbol{\lambda}'$

---

**Algorithm 3** Wasserstein Aggregation Domain Network (Limited Target Data, one iteration)

---

**Require:** Labeled source samples $\hat{\mathcal{S}}_1, \ldots, \hat{\mathcal{S}}_T$, Target samples $\hat{\mathcal{T}}$, Label shift ratio $\alpha_t$
**Ensure:** Task relation simplex $\boldsymbol{\lambda}$. Feature Learner $g$, Classifier $h$, Statistic critic function $d_1, \ldots, d_T$, class centroid for source $\mathbf{C}_t^y$ and target $\mathbf{C}^y$ ($\forall t = [1, T], y \in \mathcal{Y}$).
  1: $\triangleright \triangleright \triangleright$ DNN Parameter Training Stage (fixed $\boldsymbol{\lambda}$) $\triangleleft \triangleleft \triangleleft$
  2: **for** mini-batch of samples $(\mathbf{x}_{\mathcal{S}_1}, \mathbf{y}_{\mathcal{S}_1}) \sim \hat{\mathcal{S}}_1, \ldots, (\mathbf{x}_{\mathcal{S}_T}, \mathbf{y}_{\mathcal{S}_T}) \sim \hat{\mathcal{S}}_T, (\mathbf{x}_{\mathcal{T}}) \sim \hat{\mathcal{T}}$ **do**
  3:      Compute the *batched* class centroid for source $C_t^y$ and target $C^y$.
  4:      Moving Average for update source/target class centroid: (We set $\epsilon_1 = 0.7$)
  5:          Source class centroid update     $\mathbf{C}_t^y = \epsilon_1 \times \mathbf{C}_t^y + (1 - \epsilon_1) \times C_t^y$
  6:          Target class centroid update     $\mathbf{C}^y = \epsilon_1 \times \mathbf{C}^y + (1 - \epsilon_1) \times C^y$
  7:      Updating $g, h, d_1, \ldots, d_T$ (SGD and Gradient Reversal), based on Eq.(6).
  8: **end for**
  9: $\triangleright \triangleright \triangleright$ Estimation $\boldsymbol{\lambda}$ $\triangleleft \triangleleft \triangleleft$
10: Solve $\boldsymbol{\lambda}$ by Sec. 2.3. (denoted as $\boldsymbol{\lambda}'$)
11: Updating $\boldsymbol{\lambda}$ by moving average: $\boldsymbol{\lambda} = \epsilon_1 \times \boldsymbol{\lambda} + (1 - \epsilon_1) \times \boldsymbol{\lambda}'$

---

# M   DATASET DESCRIPTION AND EXPERIMENTAL DETAILS

## M.1   AMAZON REVIEW DATASET

We used the amazon review dataset (Blitzer et al., 2007). It contains four domains (Books, DVD, Electronics, and Kitchen) with positive (label "1") and negative product reviews (label "0"). The data size is 6465 (Books), 5586 (DVD), 7681 (Electronics), and 7945 (Kitchen). We follow the common data pre-processing strategies Chen et al. (2012): use the bag-of-words (BOW) features then extract the top-5000 frequent unigram and bigrams of all the reviews.

We also noticed the original data-set are label balanced $\mathcal{D}(y = 0) = \mathcal{D}(y = 1)$. To enhance the benefits of the proposed approach, we create a new dataset with label distribution drift. Specifically, in the experimental settings, we randomly drop $50\%$ data with label "0" (negative reviews) for all the source data while keeping the target identical, showing in Fig (5).

We choose the MLP model with

- feature representation function $g$: $[5000, 1000]$ units
- Task prediction and domain discriminator function $[1000, 500, 100]$ units,

We choose the dropout rate as $0.7$ in the hidden and input layers. The hyper-parameters are chosen based on cross-validation. The neural network is trained for $50$ epochs and the mini-batch size is $20$ per domain. The optimizer is Adadelta with a learning rate of $0.5$.

**Experimental Setting** We use the amazon Review dataset for two transfer learning scenarios (limited target labels and unsupervised DA). We first randomly select 2K samples for each domain. Then we create a drifted distribution of each source, making each source $\approx 1500$ and target sample still 2K.

In the unsupervised DA, we use these labeled source tasks and *unlabelled* target task, which aims to predict the labels on the target domain.

In the conventional transfer learning, we random sample only $10\%$ dataset ($\approx 200$ samples) as the target training set and the rest $90\%$ samples as the target test set.

We select $C_0 = 0.01$ and $C_1 = 1$ for these two transfer scenarios. In both practical settings, we set the maximum training epoch as 50.

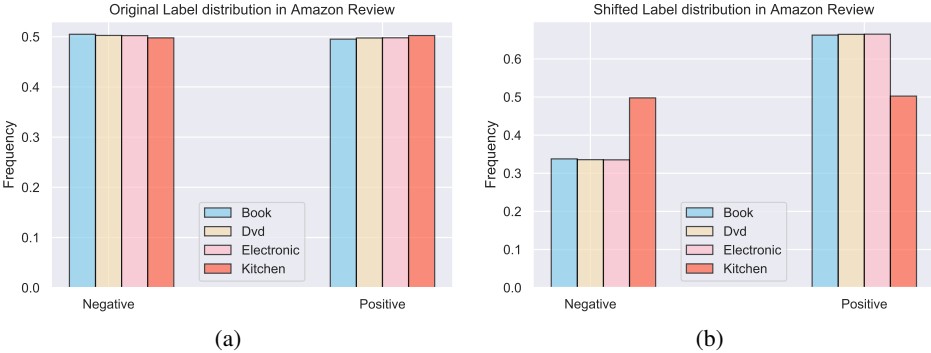

Figure 5: Amazon Review dataset (a) Original Label Training Distribution; (b) Label-Shifted distribution with sources tasks: Book, Dvd, Electronic, and target task Kitchen. We randomly drop $50\%$ of the negative reviews for all the source distribution while keeping the target label distribution unchanged.

## M.2 DIGIT RECOGNITION

We follow the same settings of Ganin et al. (2016) and we use four-digit recognition datasets in the experiments MNIST, USPS, SVHN, and Synth. MNIST and USPS are the standard digits recognition task. Street View House Number (SVHN) Ganin et al. (2016) is the digit recognition dataset from house numbers in Google Street View Images. Synthetic Digits (Synth) Ganin et al. (2016) is a synthetic dataset that by various transforming SVHN dataset.

We also visualize the label distribution in these four datasets. The original datasets show an almost uniform label distribution on the MNIST as well as Synth, (showing in Fig. 7 (a)). In our paper, we generate a label distribution drift on the source datasets for each multi-source transfer learning. Concretely, we drop $50\%$ of the data on digits 5-9 of all the sources while we keep the target label distribution unchanged. (Fig. 7 (b) illustrated one example with sources: Mnist, USPS, SVHN, and Target Synth. We drop the labels only on the sources.)

MNIST and USPS images are resized to $32 \times 32$ and represented as 3-channel color images to match the shape of the other three datasets. Each domain has its own given training and test sets when downloaded. Their respective training sample sizes are 60000, 7219, 73257, 479400, and the respective test sample sizes are 10000, 2017, 26032, 9553.

The model structure is shown in Fig. 6. There is no dropout and the hyperparameters are chosen based on cross-validation. It is trained for 60 epochs and the mini-batch size is 128 per domain. The optimizer is Adadelta with a learning rate of 1.0. We adopted $\gamma = 0.5$ for MDAN and $\gamma = 0.1$ for DARN in the baseline (Wen et al., 2020).

**Experimental Setting**    We use the Digits dataset for two transfer learning scenarios (limited target labels and unsupervised DA). Notice the USPS data has only 7219 samples and the digits dataset is relatively simple. We first randomly select 7K samples for each domain. We create a drifted distribution of each source, making each source $\approx 5300$, and the target sample still 7K.

In the unsupervised DA, we use these labeled source tasks and *unlabelled* target task, which aims to predict the labels on the target domain.

In the transfer learning with limited data, we random sample only $10\%$ dataset ($\approx 700$ samples) as the target training set and the rest $90\%$ samples as the target test set.

We select $C_0 = 0.01$ and $C_1$ as the maximum prediction loss $C_1 = \max_t R^{\alpha_t}(h)$ as the hyper-parameters across these two scenarios. The maximum training epoch is 60.

1. Feature extractor: with 3 convolution layers.
   'layer1': 'conv': [3, 3, 64], 'relu': [], 'maxpool': [2, 2, 0],
   'layer2': 'conv': [3, 3, 128], 'relu': [], 'maxpool': [2, 2, 0],
   'layer3': 'conv': [3, 3, 256], 'relu': [], 'maxpool': [2, 2, 0],

2. Task prediction: with 3 fully connected layers.
   'layer1': 'fc': [*, 512], 'act_fn': 'relu',
   'layer2': 'fc': [512, 100], 'act_fn': 'relu',
   'layer3': 'fc': [100, 10],

3. Domain Discriminator: with 2 fully connected layers.
   *reverse_gradient*()
   'layer1': 'fc': [*, 256], 'act_fn': 'relu',
   'layer2': 'fc': [256, 1],

Figure 6: Neural Network Structure in the digits recognition (Ganin et al., 2016)

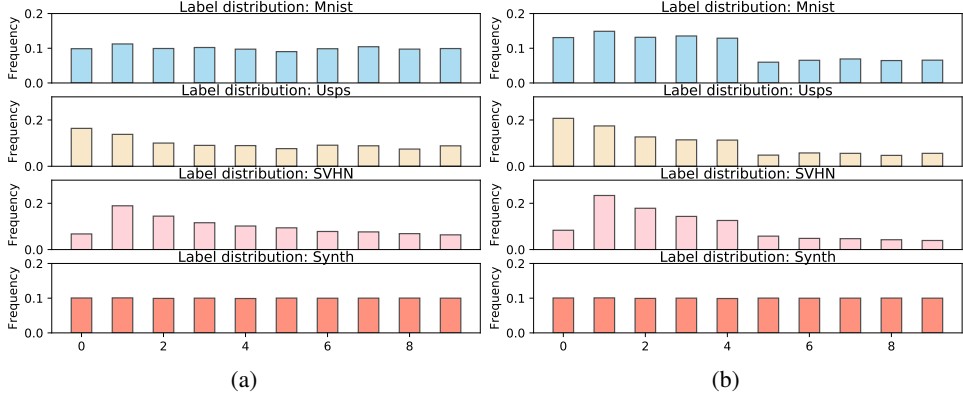

(a)                                          (b)

Figure 7: One example in Digits dataset with Sources: MNIST, USPS, SVHN and Target Synth. We randomly drop $50\%$ data on digits 5-9 in all sources while keeping target label distribution unchanged.

## M.3    OFFICE-HOME DATASET

To show the dataset in the complex scenarios, we use the challenging Office-Home dataset (Venkateswara et al., 2017). It contains images of 65 objects such as a spoon, sink, mug, and

pen from four different domains: Art (paintings, sketches, and/or artistic depictions), Clipart (clipart images), Product (images without background), and Real-World (regular images captured with a camera). One of the four datasets is chosen as an unlabelled target domain and the other three datasets are used as labeled source domains.

The dataset size is 2427 (Art), 4365 (Clipart), 4439 (Product), 4357 (Real-World). We follow the same training/test procedure as (Wen et al., 2020). We additionally visualize the label distribution $\mathcal{D}(y)$ in four domains in Fig.9, which illustrated the inherent different label distributions. We did not re-sample the source label distribution to uniform distribution in the data pre-processing step. All the baselines are evaluated under the same setting.

We use the ResNet50 (He et al., 2016) pretrained from the ImageNet in PyTorch as the base network for feature learning and put an MLP with the network structure shown in Fig. 10.

**Experimental Settings**  We use the original Office-Home dataset for two transfer learning scenarios (unsupervised DA and label-partial unsupervised DA). We use SGD optimizer with learning rate 0.005, momentum 0.9 and weight_decay value 1e-3. It is trained for 100 epochs and the mini-batch size is 32 per domain. As for the baselines, MDAN use $\gamma = 1.0$ while DARN use $\gamma = 0.5$. We select $C_0 = 0.01$ and $C_1$ as the maximum prediction loss $C_1 = \max_t R^{\alpha_t}(h)$ as the hyper-parameters across these two scenarios.

In the multi-source unsupervised partial DA, we randomly select 35 classes from the target (by repeating 3 samplings), then at each sampling we run 5 times. The final result is based on these $3 \times 5 = 15$ repetitions.

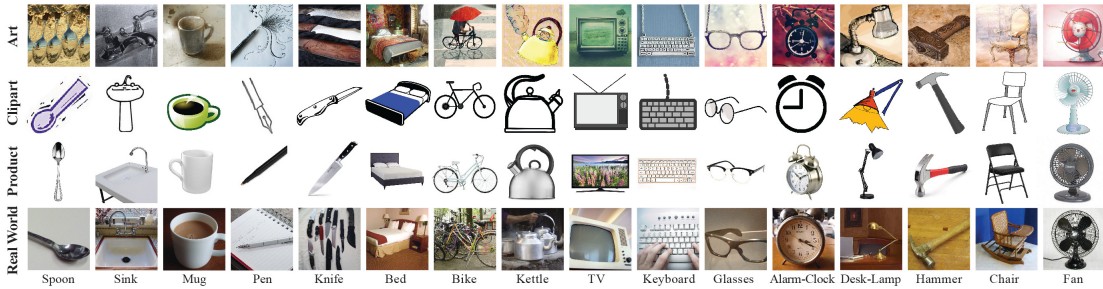

Figure 8: Samples Images From Office-Home dataset (Venkateswara et al., 2017), which consists four domains with non-uniform label distribution.

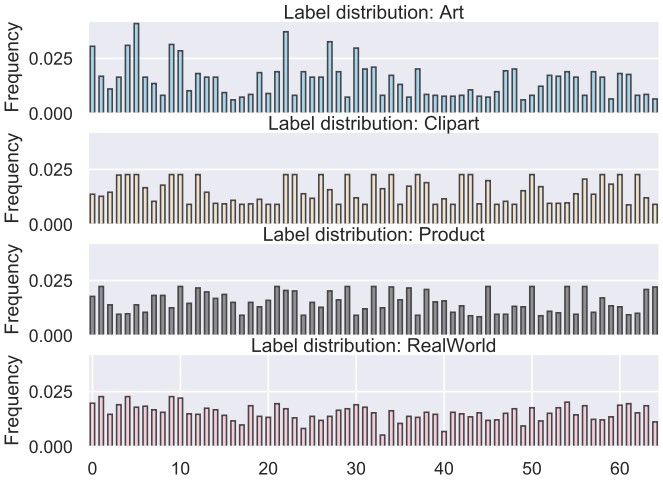

Figure 9: Label distribution of Office-Home Dataset

# N  ANALYSIS ON THE PSEUDO-LABELS

1. Feature extractor: ResNet50 (He et al., 2016),
2. Task prediction: with 3 fully connected layers.
   'layer1': 'fc': [*, 256], 'batch_normalization', 'act_fn': 'Leaky_relu',
   'layer2': 'fc': [256, 256], 'batch_normalization', 'act_fn': 'Leaky_relu',
   'layer3': 'fc': [256, 65],
3. Domain Discriminator: with 3 fully connected layers.
   *reverse_gradient*()
   'layer1': 'fc': [*, 256], 'batch_normalization', 'act_fn': 'Leaky_relu',
   'layer2': 'fc': [256, 256], 'batch_normalization', 'act_fn': 'Leaky_relu',
   'layer3': 'fc': [256, 1], 'Sigmoid',

Figure 10: Neural Network Structure in the Office-Home

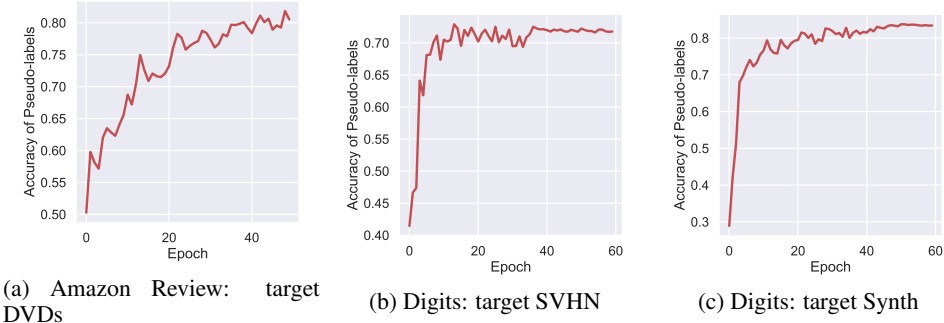

(a) Amazon Review: target DVDs

(b) Digits: target SVHN

(c) Digits: target Synth

Figure 11: Evolution of accuracy w.r.t. the predicted target pseudo-labels in different tasks in unsupervised DA.

## O  ANALYSIS IN UNSUPERVISED DA

### O.1  ABLATION STUDY: DIFFERENT DROPPING RATE

To show the effectiveness of our proposed approach, we change the drop rate of the source domains, showing in Fig.(12). We observe that in task Book, DVD, Electronic, and Kitchen, the results are significantly better under a large label-shift. In the initialization with almost no label shift, the state-of-the-art DARN illustrates a slightly better ($< 1\%$) result.

### O.2  ADDITIONAL ANALYSIS ON AMAZON DATASET

We present two additional results to illustrate the working principles of WADN, showing in Fig. (13) and (14).

We visualize the evolution of $\lambda$ between DARN and WADN, which both used theoretical principled approach to estimate $\lambda$. We observe that in the source shifted data, DARN shows an inconsistent estimator of $\lambda$ Fig. (13). This is different from the observation of Wen et al. (2020). We think it may in the conditional and label distribution shift problem, using $\hat{R}_{\mathcal{S}}(h(z)) + \text{Discrepancy}(\mathcal{S}(z), \mathcal{T}(z))$ to update $\lambda$ is unstable. In contrast, WADN illustrates a relative consistent estimator of $\lambda$ under the source shifted data.

In addition, WARN gradually and correctly estimates the unbalanced source data and assign higher wights $\alpha_t$ for label $y = 0$ (first row of Fig.(14)). These principles in WADN jointly promote significantly better results.

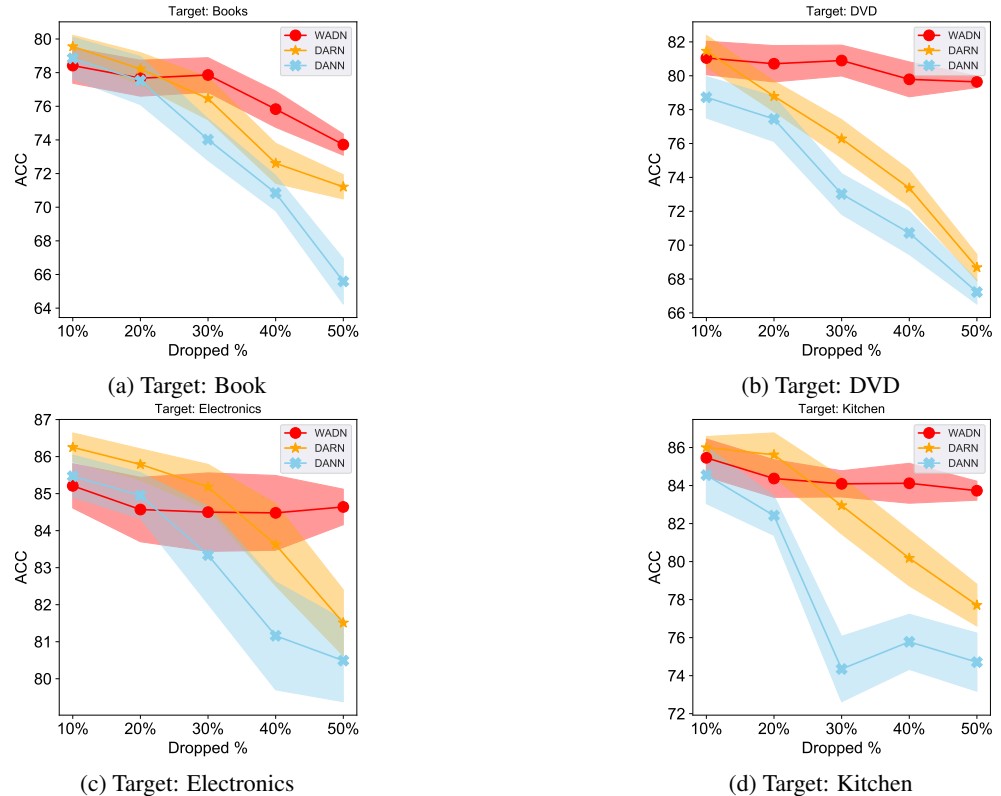

Figure 12: Different label drift levels on Amazon Dataset. Larger dropping rate means higher label shift.

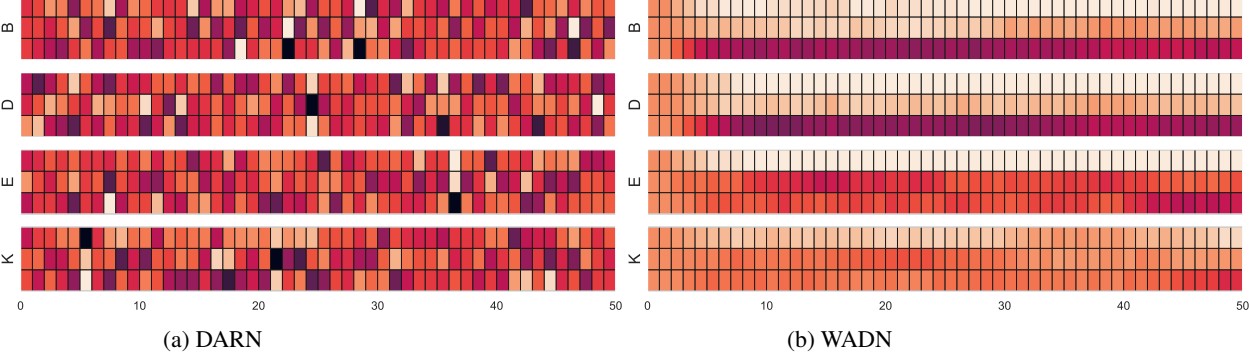

Figure 13: Source Shifted Amazon Dataset. Evolution of $\boldsymbol{\lambda}$ during the training. B=Books, D=DVD, E=Electronics, K=Kitchen.

### O.3 ADDITIONAL ANALYSIS ON DIGITS DATASET

We show the evolution of $\hat{\alpha}_t$ on WADN, which verifies the correctness of our proposed principle. Since we drop digits 5-9 in the source domains, the results in Fig. (15) illustrate a higher $\hat{\alpha}_t$ on these digits.

## P PARTIAL MULTI-SOURCE UNSUPERVISED DA

From Fig. (16), WADN is consistently better than other baselines, given different selected classes.

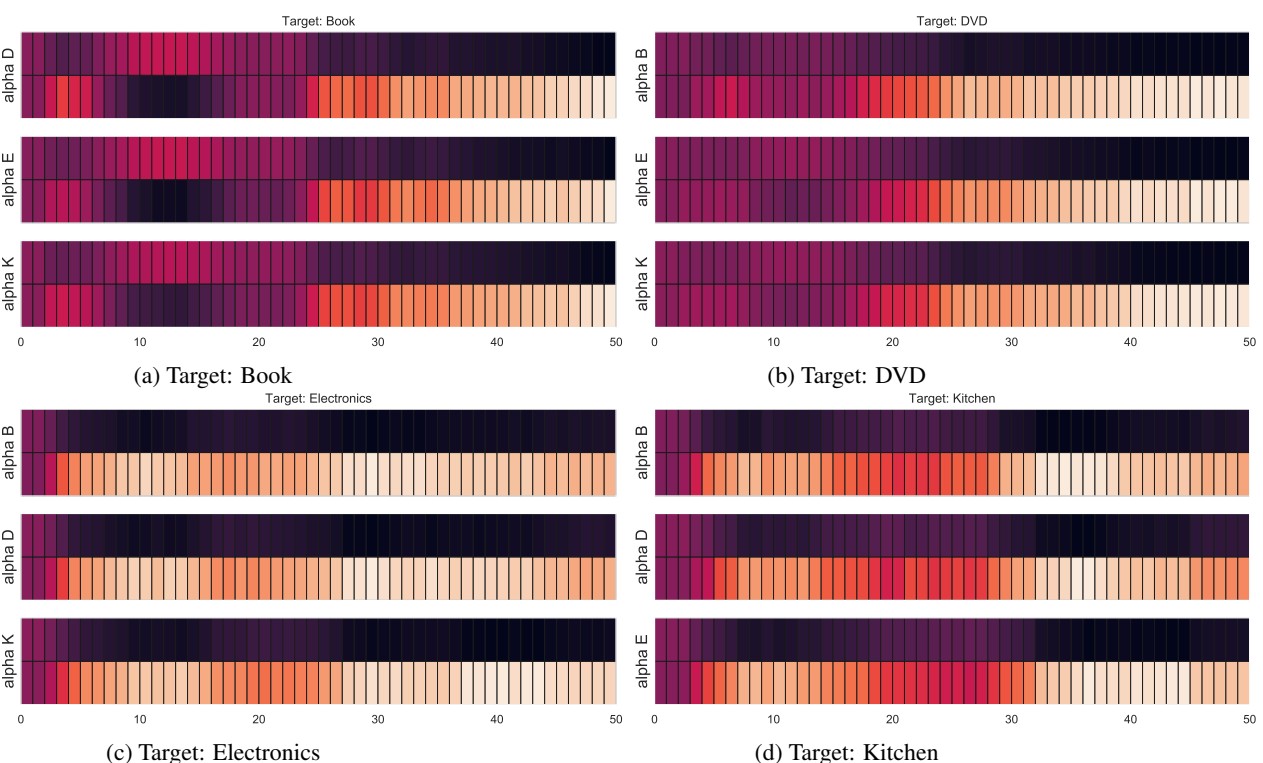

Figure 14: Amazon Dataset. WADN approach: evolution of $\hat{\alpha}_t$ during the training. Darker indicates higher Value. Since we drop $y = 0$ in the sources, then the true $\alpha_t(0) > 1$ will be assigned with higher value.

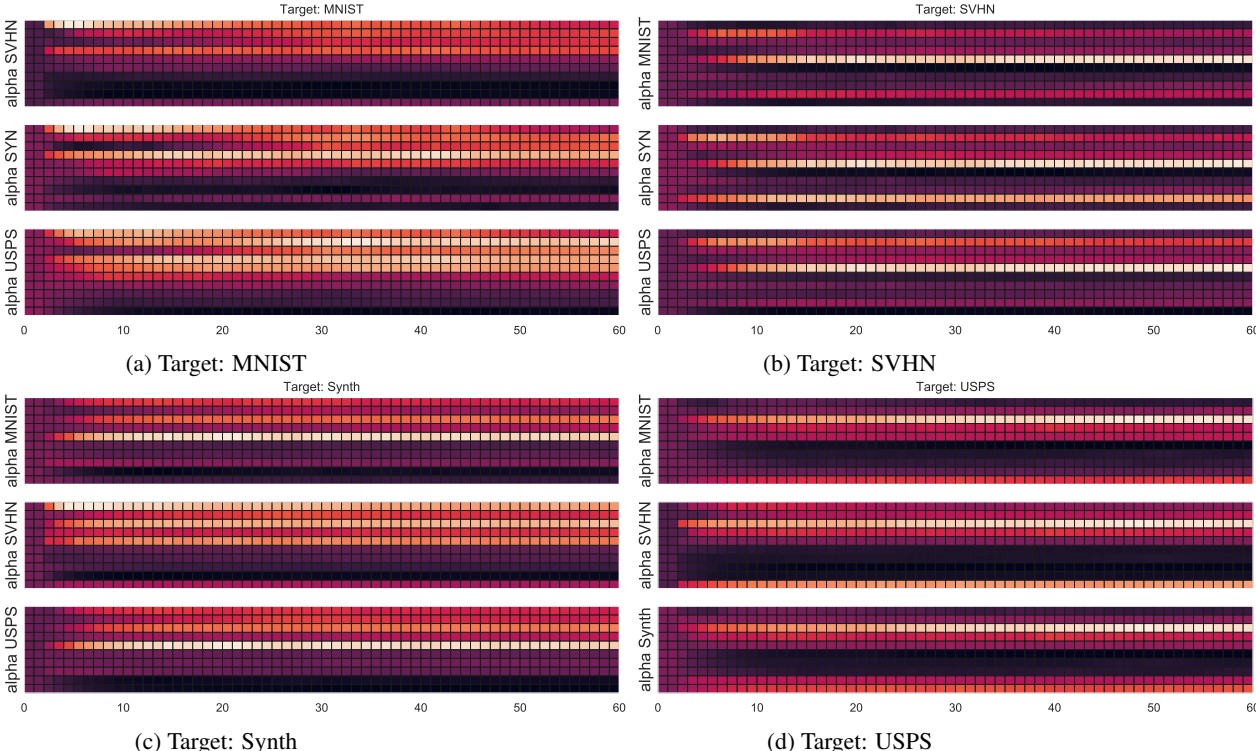

Figure 15: Digits Dataset. WADN approach: evolution of $\hat{\alpha}_t$ during the training. Darker indicates higher value. Since we drop digits $5 - 9$ on source domain, therefore, $\alpha_t(y)$, $y \in [5, 9]$ will be assigned with a relative higher value.

Besides, when fewer classes are selected, the accuracy in DANN, PADA, and DARN is not drastically dropping but maintaining a relatively stable result. We think the following possible reasons:

- The reported performances are based on the **average of different selected sub-classes rather than one sub-class selection.** From the statistical perspective, if we take a close look at the **variance**, the results in DANN are *much more unstable* (higher std) inducing by the different samplings. Therefore, the conventional domain adversarial training is improper for handling the partial transfer since it is not reliable and negative transfer still occurs.

- In multi-source DA, it is equally important to detect the non-overlapping classes and find the most similar sources. Comparing the baselines that only focus on one or two principles shows the importance of unified principles in multi-source partial DA.

- We also observe that in the Real-World dataset, the DANN improves the performance by a relatively large value. This is due to the inherent difficultly of the learning task itself. In fact, the Real-World domain illustrates a much higher performance compared with other domains. According to the Fano lower bound, *a task with smaller classes is generally easy to learn*. It is possible the vanilla approach showed improvement but still with a much higher variance.

Fig (17), (18) showed the estimated $\hat{\alpha}_t$ with different selected classes. The results validate the correctness of WADN in estimating the label distribution ratio.

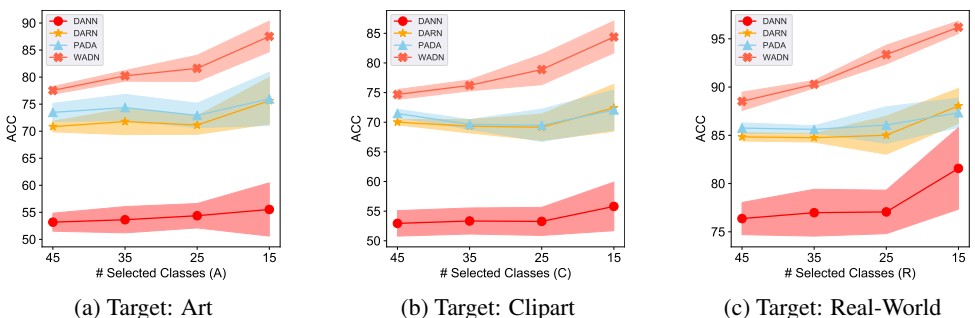

(a) Target: Art  (b) Target: Clipart  (c) Target: Real-World

Figure 16: Multi-source Label Partial DA: Performance with different target selected classes.

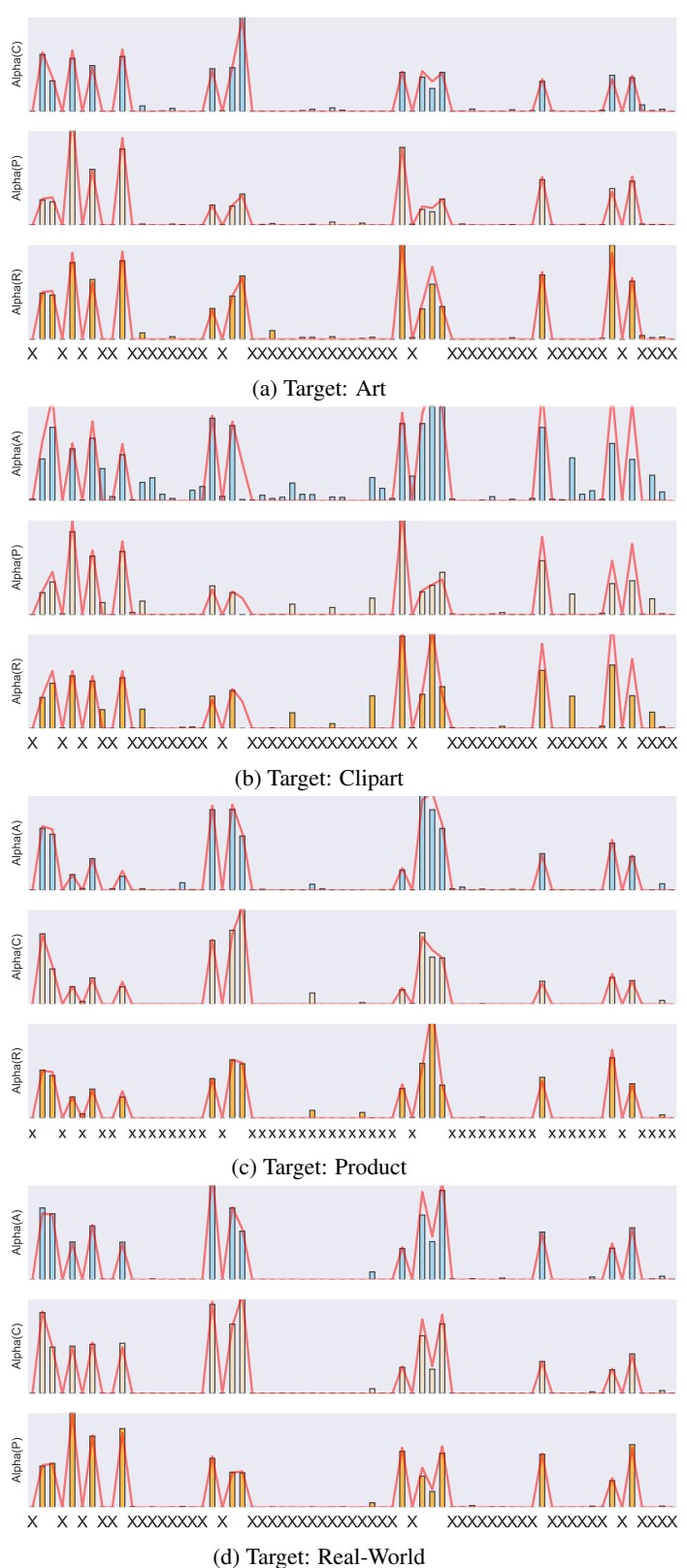

Figure 17: We select 15 classes and visualize estimated $\hat{\alpha}_t$ (the bar plot). The "X" along the x-axis represents the index of **dropped** 50 classes. The red curves are the ground-truth label distribution ratio.

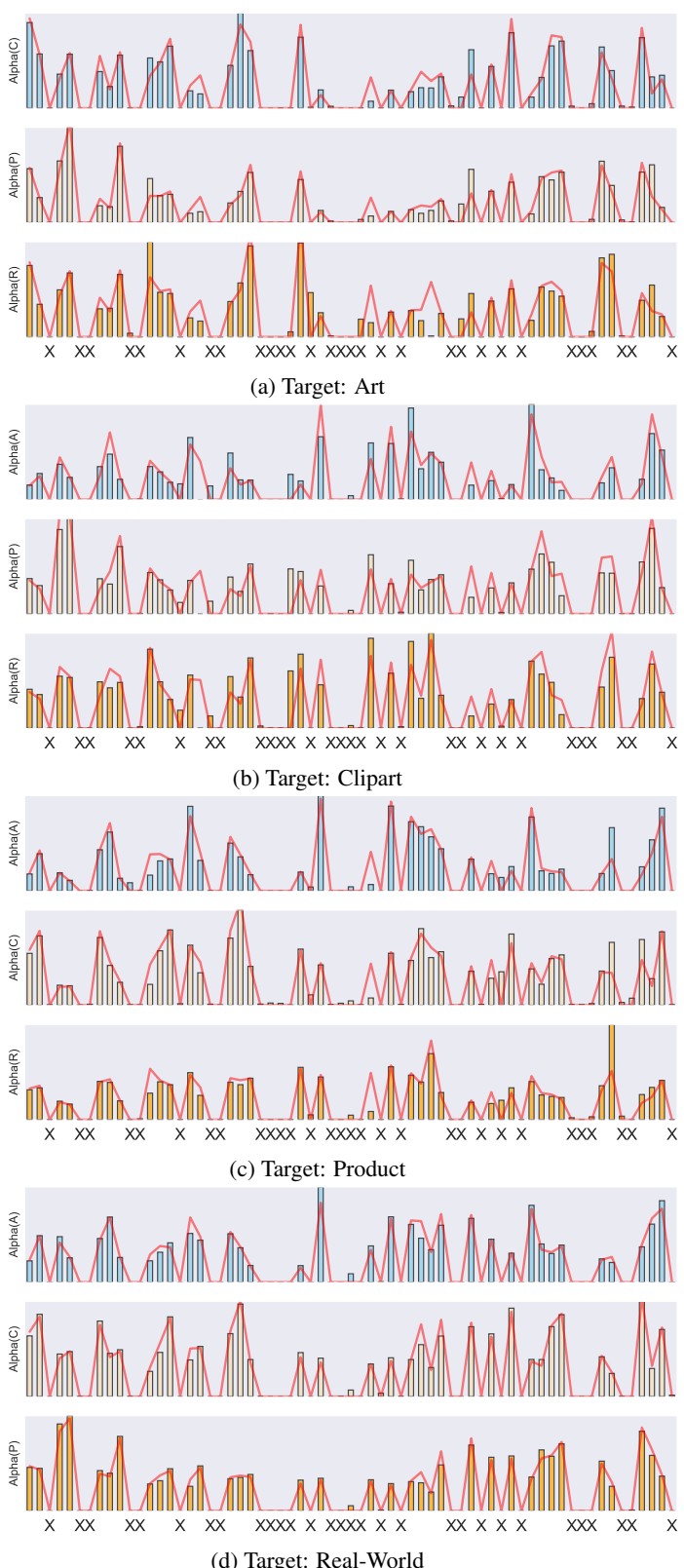

Figure 18: We select 35 classes and visualize estimated $\hat{\alpha}_t$ (the bar plot). The "X" along the x-axis represents the index of **dropped** 30 classes. The red curves are the ground-truth label distribution ratio.

