# OpenReview forum: "Unified Principles For Multi-Source Transfer Learning Under Label Shifts"
_ICLR.cc/2021/Conference — Reject_

### Official Review · AnonReviewer1 · 2020-10-25
**Interesting and potentially impactful, but raw/strangely organized/hard to follow paper**

**Rating:** 7
**Confidence:** 4

**Review:**

Summary:
The paper is concerned with label shift in multi-source transfer learning setup. In particular authors look into target shift without assuming conditional distributions to be the same in source and target. They propose a unified frameworkthat can be used for learning with no target labels and limited target labels.
They show that the performance on target depends on how well we are able to estimate the ratio of label distrib between source and target, and the gap between real and estimated ratios,. It also depends on the weights assigned to each source tasks. Authors then proceed to show how to learn a model in various settings using the knowledge about the bounds.

Overall, this paper CAN be really impactful, but as of now, it is really hard to follow and understand how it compares to other methods (not just empirically).
1) The structure of the paper is strange, it is impossible to just read the paper and get enough information on any of the subsections,  everything is in the appendix. Related work is missing (see appendix), the theorem with bounds (that is used through the paper) appears without at least a sketch/idea of how authors got to it and where wasserstein distance comes from. Since it is not intuitive, and since it is not clear how these bounds compare to DANN bounds (https://arxiv.org/pdf/1505.07818.pdf or Ben-David bounds for that matter), it is really hard to believe in the algorithm authors derive, even though it seems to give good empirical improvements
Also K section in appendix is essential for understanding and should be moved to the main paper.
2) Related work absolutely needs to be in the main paper It is impossible to understand what connection your work has with respect to established models like DANN and CDANN. Additionally Wassertein distance was used in a number of papers already,  https://arxiv.org/pdf/2009.02831.pdf, https://arxiv.org/pdf/1707.01217.pdf, https://arxiv.org/pdf/1909.08675.pdf. Again not clear on connection if any and comparison
3) Experiments:
- For DANN, did i understand correctly that you take all source tasks as one "source" domain and all the target as another domain. Why not to consider all the source tasks as separate domains and target as an additional domain, DANN allows for it easily (CE instead of log loss in adv head)
- You should be comparing to CDAN https://arxiv.org/pdf/1705.10667.pdf which has been shown to better than DANN
-  Also didn't understand your setup for experiments. For example on digits, what are your sources and what is your target task? A table shows only target, does it mean that source is all other digits datasets?

Minor:
- in Into, you mention "without assuming that conditional distributions are the same". I think it is most common to assume p(y|x) is the same, not p(x|y) as you show.
- what exactly is "label partial unsupervised domain adaptation"

---

> ### Author Response · Authors · 2020-11-18
> **Response 1/3**
>
> Thanks for your constructive feedback and additional references!
>
> Due to the strict 8-pages limit during the submission, we have to put many materials in the appendix. We apologize for the inconvenience in understanding the paper. We notice the main concern is about the structure and organization of the paper, which has been fixed in our revised version. We hope the following responses and our revised paper can help you have a better understanding of our submission.  If you have additional questions, we are happy to provide more explanations!
>
> #### Q1
>
> >Paper Structure
>
> Thanks for your suggestion and we have carefully revised our paper accordingly.
>
> -- We moved the related work in Sec 2.
>
> -- We added more insights and discussions about our theoretical results in Sec.3
>
> -- We added a full algorithm description in the proposed algorithm.
>
> > How these bounds compare to previous (such as DANN) bounds
>
> We notice that H-divergence [1] or discrepancy distance [2] are well established in multi-source DA theory, with the following general form:
>
> $$R_T(h) \leq \sum_{t}\lambda[t] R_S(h) + \sum_{t}\lambda[t] d(S(x),T(x)) + \eta$$
>
> Where $\eta$ is the optimal risk on all the source and target domains and unobservable in the UDA.
>
> * However, [3,4] pointed out that **merely minimizing the first two terms will be problematic when label shift occurs.** It will lead to an increase in $\eta$, which means the upper bound is not necessarily small by minimizing the first two terms (See Theorem 4.3 in [4] for details.)
>
> * [5-8] additionally mentioned that $\eta$ is theoretically impossible to bound in UDA. To address this, Ref. [2] pointed out that in order to theoretically take $\eta$ into consideration in label shifts with different conditional distributions, a small amount of target labelled data is necessary. Based on this, *we proved an alternative bound without $\eta$.*
>
> * Besides, we explicitly proposed a theoretical analysis in the context of representation (Theorem 2), which enables a clear guideline in learning the representation function $g$ w.r.t. the latent space $z$. This remains elusive from previous theorietical analysis in multi-source DA [1-2], which did not analyze the DA in the feature space $z$.
>
> > Where Wasserstein distance comes from (Proof idea by using Wasserstein-Distance)
>
> * Thanks for your comments. We used Wasserstein distance since it is theoretically tighter than other commonly used distances such as TV distance [1] or Jensen-Shannon divergence [3]. Therefore using Wasserstein distance can recover the previous theoretical results. In addition, Wasserstein distance takes into account the geometrical information of the distribution, which can be advantageous over other divergences [12].
>
> * In our proof, the Wasserstein distance is introduced through the duality theorem of Kantorovich and Rubinstein. i.e If f(x) is K-Lipschitz, then we have:
>
> $$E_{x \sim P} f(x) - E_{x\sim Q} f(x) \leq K W_1(P(x)||Q(x))$$
>
> Since we assume the loss and classifier are Lipschitz functions, we can use this inequality to derive the theoretical results with Wasserstein distance. Moreover, the Lipschitz property is a reasonable assumption in a deep neural-network, where the Lipschitz properly can be approximated and enforced through recent works [13-15].
>
> #### Q2  Discussion with Related work.
> Thanks for your suggestions. We have revised our related work in our paper. The following are the additional discussions between our paper and suggested papers.
>
> > Connection your work has with respect to established models like DANN and CDANN.
>  Wasserstein distance was used in a number of papers already
>
> *  DANN/CDAN/WDANN [9-11]. These papers all talked about *one source* UDA problem. In contrast, our paper aims at solving *multiple sources* to one target transfer problem, which is more challenging. As [1,2,12] highlights the importance of aggregating different sources for avoiding a negative transfer. These single transfer learning papers did not propose guidelines of how to rank its task relatedness.
>
> * **Besides, compared with Wasserstein distance-based DA papers [9-11], our proposed principles are different. [9-11] still aims at minimizing $R_S(h) + W1(S(z), T(z))$ without considering the influence of $\eta$**, which will suffer a poor performance when label-shift occurs`.
>
> * Moreover, we empirically compared a SOTA in multi-source DA using Wasserstein distance (MDMN, [12]). Our empirical evidence suggests the strong benefits of our proposed principles under Wasserstein distance, because of the explicit approach for handling label shift.

---

> > ### Author Response · Authors · 2020-11-18
> > **Response 2/3**
> >
> > #### Q3 Experiments
> >
> > >  Why not to consider all the source tasks as separate domains and target as an additional domain, DANN allows for it easily (CE instead of log loss in adv head)
> >
> > Thanks for your suggestion. We follow your idea and add the experiments in the Office-Home dataset, shown in Tab.1
> >
> > #### Tab.1 Modified DANN on Office-Home dataset (accuracy %)
> >
> > |Target | Art | Clipart | Product | RealWorld |Average|
> > |---------|--------------------|----------------------------|------------------|---------|----------|
> > |Modified-DANN| 65.89 $\pm$ 0.34 |60.75 $\pm$ 0.57 |70.12 $\pm$ 0.52 |72.86 $\pm$ 0.66 | 67.41 |
> > |Ours |73.78 $\pm$ 0.43 | 70.18 $\pm$ 0.54 | 86.32 $\pm$ 0.38 | 87.28 $\pm$ 0.87 | 79.39 |
> >
> > The empirical result indicates our approach is still significantly better than modified DANN. We think the main reasons are in the multi-source DA, each source is not equally important for the target domain. Then the negative transfer can occur without exploring the relatedness of each source.
> >
> >
> > > You should be comparing to CDAN
> >
> > Thanks for your suggestion. We follow your suggestion and add experiments of CDAN in the Office-Home dataset. Since the original idea of CDAN did not provide a guideline on how to learn multiple sources, we still merge all sources as a big source domain.
> >
> >
> >
> > #### Tab.2 CDAN on Office-Home dataset (accuracy %)
> >
> > |Target | Art | Clipart | Product | RealWorld |Average|
> > |---------|--------------------|----------------------------|------------------|---------|----------|
> > |CDAN| 69.21 $\pm$ 0.61 | 67.81 $\pm$ 0.31 | 78.36 $\pm$ 0.68 | 79.34 $\pm$ 0.49 | 73.68|
> > |Ours |73.78 $\pm$ 0.43 | 70.18 $\pm$ 0.54 | 86.32 $\pm$ 0.37 | 87.28 $\pm$ 0.87 | 79.39 |
> >
> > The empirical result indicates our approach is significantly better than CDAN. We think the reasons are (a) CDAN did not propose the mechanism for handling label shift, shown in [3]; (b) CDAN did not use the relateless of tasks.
> >
> > >.... on digits, what are your sources and what is your target task? A table shows only target, does it mean that source is all other digits datasets?
> >
> > Thanks for your remark. Your understanding is correct. The target indicates the target domain and we set all other three datasets as sources. e.g In office-home If the target is Art, the sources are Clipart, Product, and Realworld.
> >
> >
> > #### Minor
> >
> > * Definition of the conditional distribution. We agree with this confusion, in our paper we refer to the *conditional distribution as semantic conditional distribution* P(x|y), which is coherent with [3,17-18]. We have revised it as a semantic conditional distribution.
> >
> > >  what exactly is "label partial unsupervised domain adaptation"
> >
> > Label partial unsupervised domain adaptation can be viewed as a special case of label shift unsupervised DA, where the label space in the target domain is a subset of the source domain (i.e., for some classes $\mathcal{S}(Y=c)\neq 0$; $\mathcal{T}(Y=c)= 0$).  For example, in digits recognition,
> >
> > -- source domain is MNIST with classifying digits 0-9.
> >
> > -- The target domain is USPS with *classifying a subset of source classes* (e.g digits classification 0-5, or 1-6, etc..).
> >
> > *Moreover, we only have the target data (x) without knowing the exact overlapped class.*
> >
> > Therefore this is much more challenging compared with multi-source UDA since it requires detecting the absent classes in the target domain and conduct a correct transfer on each class. Our proposed principle can exactly detect the absent classes, which verifies the benefits of our principles (Fig 2(b)).

---

> > > ### Author Response · Authors · 2020-11-18
> > > **Response 3/3**
> > >
> > > #### References:
> > >
> > > [1] Adversarial multiple source domain adaptation. Zhao, et.al NeurIPS 2018
> > >
> > > [2] Domain aggregation networks for multi- source domain adaptation. Wen, et.al ICML 2020
> > >
> > > [3] Domain adaptation with conditional distribution matching and generalized label shift Combes, et.al. 2020
> > >
> > > [4] On Learning Invariant Representation for Domain Adaptation Zhao, et.al ICML 2019
> > >
> > > [5] Robust Learning from Untrusted Sources. Konstantinov.et.al ICML 2019
> > >
> > > [6] On the Value of Target Data in Transfer Learning. Hanneke.et.al NeurIPs 2019
> > >
> > > [7] Transfer Learning via Minimizing the Performance Gap Between Domains Wang.et.al NeurIPS 2019
> > >
> > > [8] A theory of multiple-source adaptation with limited target labeled data. Mansour,et.al 2020
> > >
> > > [9] Wasserstein Distance Guided Representation Learning for Domain Adaptation. Shen AAAI, 2018
> > >
> > > [10] Wasserstein distance based domain adaptation for object detection. Xu, 2019
> > >
> > > [11] Unsupervised Wasserstein Distance Guided Domain Adaptation for 3D Multi-Domain Liver Segmentation You, 2020
> > >
> > > [12] Extracting relationships by multi-domain matching. Li, et.al NeurIPS 2018
> > >
> > > [13] Lipschitz regularity of deep neural networks: analysis and efficient estimation. Scaman, et.al NeurIPS 2018
> > >
> > > [14] Efficient and Accurate Estimation of Lipschitz Constants for Deep Neural Networks. Fazlyab, et.al NeurIPS 2019
> > >
> > > [15] Regularisation of Neural Networks by Enforcing Lipschitz Continuity. Gouk, et.al 2018
> > >
> > > [16]Theoretical Analysis of Domain Adaptation with Optimal Transport. Redeo, et.al ECML 2017
> > >
> > > [17] Domain adaptation with conditional transferable components. Gong ICML 2016
> > >
> > > [18] Domain adaptation under target and conditional shift. Zhang, ICML 2013

---

> > > > ### Author Response · Authors · 2020-11-21
> > > > **Thanks for your addtional comments.**
> > > >
> > > > [Since the OpenReview system does not support the direct message to the individual reviewer, we post our additional response in a new comment]
> > > >
> > > > Thanks for your constructive feedback and comments! These indeed help improve the readability and quality of our paper.
> > > >
> > > > We currently start running new experiments (modified CDAN), as you suggested on the Office-Home dataset. We are trying to finish these experiments before the final deadline for updating the rebuttal/paper (11.24).
> > > >
> > > > -------Updates for modified CDAN----------
> > > >
> > > > We added new experiments (modified CDAN), as the reviewer suggested on the Office-Home dataset.
> > > >
> > > >  Tab.3 Modified-CDAN on Office-Home dataset (accuracy %)
> > > >
> > > > |Target | Art | Clipart | Product | RealWorld |Average|
> > > > |---------|--------------------|----------------------------|------------------|-----------|------------|
> > > > |Modified-CDAN| 70.81 $\pm$ 0.73 | 69.69 $\pm$ 0.45 | 83.52 $\pm$ 0.55 | 85.67 $\pm$ 0.47 | 77.42|
> > > > |Ours |73.78 $\pm$ 0.43 | 70.18 $\pm$ 0.54 | 86.32 $\pm$ 0.37 | 87.28 $\pm$ 0.87 | 79.39 |
> > > > The empirical results suggest a consistently better performance.

---

### Official Review · AnonReviewer3 · 2020-10-28
**The paper has good theoretical result supported by experiments.**

**Rating:** 6
**Confidence:** 5

**Review:**

This paper has made a good attempt to provide a unified approach for unsupervised domain adaptation. The proposed approach is applicable to three scenarios which have been traditionally  treated as three separate problems. The three problems that are treated in a unified way are Unsupervised Domain Adaptation (UDA), limited target labels and partially unsupervised domain adaptation. Another feature of the proposed approach is that it deals with target shift without assuming that conditional distributions are identical, a more realistic assumption for real-world problems.  Results on three different benchmark datasets are provided. Results show uniform improvements in the range of 2-6% over methods compared in various tables. The paper can be improved by providing comparisons with recent UDA and domain generalization methods from Balaji, Sankaranarayanan (CVPR 2018, NIPS 2018) and Balaji and Feizi (ICCV 2019). The ICCV paper also uses the Wasserstein distance for unsupervised domain adaptation. Given that one of the problems that is considered is UDA, I am not sure why the authors have not compared their approach on the Office dataset that is used in UDA papers. A recent paper from Saenko and Trevor Darrell [Saito, K., Kim, D., Sclaroff, S., Darrell, T., & Saenko, K. (2019). Semi-supervised Domain Adaptation via Minimax Entropy. arXiv preprint arXiv:1904.06487] has considered the problem of small source. small target and large unlabeled target data as a domain adaptation problem. Since the authors consider the limited target labels problem as a one of the cases, comparisons with this paper should also be provided.

---

> ### Author Response · Authors · 2020-11-18
> **Response**
>
> Thanks for your constructive comments and additional references, we hope the following responses can help you have a better understanding of our papers. If you have additional questions, we are happy to provide more explanations!
>
> #### 1. Related work in UDA.
>
> Thanks for your suggestions. We have added all the aforementioned works [1-4] in our related work as a board discussion.
>
> Since our paper aims at solving multi-source transfer learning (such as multi-source UDA, multi-source partial DA problems). As we know that to avoid a negative transfer, we need to aggregate different sources to find the most related source to conduct a transfer. The aforementioned papers [2-3] generally focused on single source DA without providing guidelines to combine multiple sources. As for domain generalization [1], since there is no target information, there may exist an adversary that can thus significantly degrade test performance by choosing P(y) with more weight on difficult classes (Ref [5]). Therefore it is difficult to conduct a fair comparison to show the good properties of such baselines.
>
> Based on these reasons, we mainly compare the paper by focusing on the multi-source DA problems (such as M3SDA, DARN, etc). We have revised our paper with additional explanations in Sec 1 and 2.
>
> #### 2. Related work in Limited target transfer learning
>
> Thanks for your suggestion. We added more experiments for MME in this setting. Since MME is originally designed for one single source DA, we simply merge all the sources as a big source dataset and conduct our experiments. The experimental settings are identical in our paper, shown in Tab 1 and 2.
>
>
>
> ##### Tab1. Accuracy on source-shifted Amazon Review dataset (accuracy %)
>
> |Target | Books | DVDs | Electronics | Kitchen | Average|
> |---------|--------------------|----------------------------|-----------------------------|-------------------|------------|
> |MME| 69.66 $\pm$ 0.58 | 71.36 $\pm$ 0.96 | 78.88 $\pm$ 1.51 | 76.64 $\pm$ 1.73 | 74.14 |
> |Ours | 74.83 $\pm$ 0.84 | 75.05 $\pm$ 0.62 | 84.23 $\pm$ 0.58 | 81.53 $\pm$ 0.90 | 78.91 |
>
>
>
>
> ##### Tab2. Accuracy on source-shifted Digits dataset (accuracy %)
>
>  |Target | Mnist | SVHN | Synth | Usps|Average |
> |---------|--------------------|----------------------------|-----------------------------|------------|-------|
> |MME| 87.24 $\pm$ 0.95 | 65.20 $\pm$ 1.35 | 80.31 $\pm$ 0.60 | 87.88 $\pm$ 0.76 | 80.16 |
> |Ours | 88.32 $\pm$ 1.17 | 70.64 $\pm$ 1.02 | 81.53 $\pm$ 1.11 | 90.53 $\pm$ 0.71 | 82.75|
>
> The empirical results indicate consistently better performance.
>
> Reference:
>
> [1] MetaReg: Towards Domain Generalization using Meta-Regularization. Bajai, et.al NeurIPS 2018
>
> [2] Normalized Wasserstein for Mixture Distributions with Applications in Adversarial Learning and Domain Adaptation. Bajai, et.al ICCV 2019
>
> [3] Generate to Adapt: Aligning Domains Using Generative Adversarial Networks. Bajai, et.al CVPR 2018
>
> [4] Semi-supervised Domain Adaptation via Minimax Entropy. Saito, K, et.al ICCV 2019
>
> [5] Coping with Label Shift via Distributionally Robust Optimisation. Zhang, et.al 2020.

---

### Official Review · AnonReviewer2 · 2020-11-02
**Unified Principles For Multi-Source Transfer Learning Under Label Shifts**

**Rating:** 7
**Confidence:** 3

**Review:**

In this paper, the authors focus on the label shift problem in multi-source transfer learning and derive new generic principles to control the target generalization risk. They propose a framework that unifies the principles of conditional feature alignment, label distribution ratio estimation, and domain relation weights estimation. A WADN algorithm is proposed for 3 multi-source label shift transfer scenarios:  learning with limited target data, unsupervised DA, and label partial unsupervised DA. The proposed WADN algorithm is validated on different scenarios on common benchmark datasets (Digits, HomeOffice, Amazon Review), and results indicate that it can outperform related SOTA methods for these scenarios.

- Although the paper is well written, and all the key concepts are described in some detail, I found some parts of Section 2 difficult to follow.  The supplementary material in appendices provides much additional information for the reader: proofs, results, etc. In terms of the organization, the authors present their review and analysis of the SOTA literature in Appendix A (no in the main paper). Therefore, it is not immediately clear in the main paper how their framework and algorithm are motivated by challenges in literature.

- The experimental validation is limited in some respects. The authors do present averaged results over independent replications, using some cross-validation process.  There should be further analysis of the impact on the performance of growing: the number and size of the sources, the degree of shift, and diversity among source and target domains? The experimental section should be expanded to compare results on different backbone networks. Their model should be also compared with SOA methods in terms of time and/or memory complexity.

- Tables 2-7 shows the lower-bound (source or source + tar), but does not show upper-bound results, like when training a DL model on the source and target data that are labeled (in the UDA scenario).  These tables also show DA results with an average value (last column). This is common in DA papers, but I still fail to see the point of averaging across different problems.

- It seems like their code is not made available, so there is a concern that the results in this paper would be very difficult for a reader to reproduce.

---

> ### Author Response · Authors · 2020-11-18
> **Response 1/2**
>
> Thanks for your feedback and constructive suggestions, we hope the following responses can help you have a better understanding of our papers. If you have additional questions, we are happy to provide more explanations!
>
> #### 1. Paper Structure
>
> Thanks for your constructive feedback. Due to the strict 8-pages limit during the submission, we have to put many materials in the appendix. We apologize for the inconvenience in understanding the paper. We have revised our paper accordingly by moving the important related work into Sec. 2 for facilitating understanding of the paper. In addition, we revised the theoretical part and provided more insights. Below we also provide a short discussion of previous works.
>
> * We aim at understanding the theory and practice of label-shift problems on multi-source transfer learning. As previous work in multi-source transfer learning (Multi-source transfer learning theory in Sec.2 of the main paper) does not consider the influence of label shift. As for the approach in Label shift (Label shift in Sec.2 of the main paper), they do not consider how to aggregate the different sources to avoid the negative transfer.
>
> * In our paper, we simultaneously consider these two aspects and derive unified principles to handle three scenarios that label shift occurs with significant empirical improvement.
>
> #### 2. Additional Empirical Validations.
> Thanks for your suggestions. We added additional experimental results as follows.
>
> >The number of sources.
>
> We added experiments in Amazon review datasets on two sources, shown in Tab 1.
> ##### Tab1. Amazon Reviews on 2 source domains (accuracy %)
>
> |Target | DVDs | Electronics | Kitchen |
> |-----------|---------------------------|-------------------------------|-----------------------------|
> |DANN | 68.62 $\pm$ 0.29 | 64.17 $\pm$ 0.49 | 74.22 $\pm$ 0.35 |
> |DARN | 69.56 $\pm$ 0.34 | 68.85 $\pm$ 0.31 | 72.43 $\pm$ 0.22 |
> |Ours | 71.17 $\pm$ 0.58 | 78.69 $\pm$ 0.37 | 75.57 $\pm$ 0.64 |
>
> The empirical results are consistently better than the most recent SOTA approach DARN.
>
> >The size of sources.
>
> We added experiments in Amazon review datasets with different sizes of sources, shown in Tab 2- Tab 3.
> ##### Tab2. Amazon Review on with source size 1000 (accuracy %)
>
> |Target | Books | DVDs | Electronics | Kitchen |
> |---------|-----------------------|---------------------------|---------------------------|-----------------------|
> |DANN | 65.09 $\pm$ 0.34| 67.17 $\pm$ 0.24 | 79.53 $\pm$ 0.19 | 74.30$\pm$ 0.44|
> |DARN | 70.02 $\pm$ 0.28| 68.57 $\pm$ 0.41 | 80.65 $\pm$ 0.24 | 77.31$\pm$ 0.33|
> |Ours | 73.10 $\pm$ 0.50 | 77.72 $\pm$ 0.55 | 83.16 $\pm$ 0.42 | 82.42$\pm$ 0.48|
>
> ##### Tab3. Amazon Review on with source size 3000 (accuracy %)
>
> |Target | Books | DVDs | Electronics | Kitchen |
> |---------|--------------------|----------------------------|-----------------------------|-------------------|
> |DANN| 70.36$\pm$ 0.33 | 72.54$\pm$ 0.38 | 81.29 $\pm$ 0.27 | 80.38$\pm$ 0.15|
> |DARN| 72.22$\pm$ 0.25 | 73.31$\pm$ 0.33 | 82.67 $\pm$ 0.15 | 82.34$\pm$ 0.24|
> |Ours | 74.78$\pm$ 0.22 | 80.30$\pm$ 0.34 | 84.11 $\pm$ 0.41 | 87.52$\pm$ 0.42|
>
> The empirical results are consistently better than the recent SOTA approach DARN under the different sizes of sources.
>
> >The degree of shift
>
> We evaluated the degree of label shift of the proposed algorithm in Fig 1(a) and Fig 12 of the paper. The ablation study indicates our proposed approaches are consistently better than the recent SOTA approach DARN under different label-shift levels.
>
> >Diversity among source and target domains
>
> Our three benchmarks indicate different diversity levels of source and target domains. For example, in office-home dataset, the four domains have different semantic information, shown in Fig 8. Besides, the digits dataset consists of two semantically different catalogs: MNIST/USPS and Svhn/Synth. Our proposed approach showed a significant improvement and a correct estimation of task relations, shown in Fig 1(b).

---

> > ### Author Response · Authors · 2020-11-18
> > **Response 2/2**
> >
> > >Comparison with SOTA with different time and memory complexity.
> >
> > Similar to (Wen ICML 2020), we discuss the time and memory complexity of our approach.
> > * Time complexity: In each mini-batch, we need to compute re-weighted losses $\mathcal{O}(T)$, domain adversarial losses $\mathcal{O}(T)$ and explicit conditional losses $\mathcal{O}(T)$. Then our computational complexity is still $\mathcal{O}(T)$ in the mini-batch level, which is comparable with SOTA such as MDAN and DARN. Then, after each training epoch we need to estimate $\alpha_t$ and $\mathbf{\lambda}$ with time complexity $\mathcal{O}(T|\mathcal{Y}|)$ , if we adopt SGD to solve these two convex problems. Therefore, the total time complexity of the proposed algorithm is $\mathcal{O}(T|\mathcal{Y}|)$. The extra $\mathcal{Y}$  is because of the approach of handling label shifts in the designed algorithm.
> > * Memory Complexity: Our proposed approach requires $\mathcal{O}(T)$-domain discriminator and $\mathcal{O}(T|\mathcal{Y}|)$ class-feature centroids. In contrast, MDAN and DARN require $\mathcal{O}(T)$ domain discriminator, and M3SDA and MDMN require $\mathcal{O}(T^2)$ domain discriminators. Since our class-feature centroids are defined in the latent space ($z$), then the memory complexity of the class-feature centroids can be much smaller than domain discriminators.
> >
> > #### 3. Upper bound and Average Values.
> >
> > * Thanks for your remarks. We have evaluated and updated our upper bound values.
> >
> > * Average values in the last column. Thanks for your comments. We agree that there is no statistical meaning to report the average on different tasks. We think the main motivation is to propose indicative information about the improved performance of the proposed approaches.
> >
> > #### 4. Code release
> >
> > Thanks for your comment. We will release the code and implementation details if the paper is accepted.
> >
> > ##### Ref:
> > [1] Domain aggregation networks for multi- source domain adaptation. Wen, et.al ICML 2020

---

### Official Review · AnonReviewer4 · 2020-11-08
**The Contribution of this paper is limited.**

**Rating:** 4
**Confidence:** 4

**Review:**

Summary
This paper aims to provide a unified principle for multi-source transfer learning under label shifts. Based on this principle, this paper claims that a unified algorithm is proposed for various multi-source label shift transfer scenarios: learning with limited target data, unsupervised domain adaptation and label partial unsupervised domain adaptation. The proposed algorithm is validated on three benchmark datasets. The proof seems correct via combining existing single-domain DA theory and the theory regarding Wasserstein distance. The main theorem (Theorem 1) assumes that we can get the label information in the target domain, which is not realistic in many DA problem settings (e.g., UDA or multi-source UDA in this paper). In many DA problem settings, we have to use pseudo labels to replace with true labels in the target domain, which should be analysed in the proposed theorem. However, this paper does not make any efforts to theoretically analyse the effect of pseudo labels, which results in that this paper has very limited impacts on the DA field. Besides, there are some misleading conclusions in this paper.

Pros:
1.	The proposed theorems seem correct.
2.	The proposed algorithm has good results after using the label-distribution ratio. The proposed method can address various DA problems and some machine learning problems.

Cons:
1.	The main theorem (Theorem 1) assumes that we can get the label information in the target domain, which is not realistic in many DA problem settings (e.g., UDA or multi-source UDA in this paper).
2.	In many DA problem settings, we have to use pseudo labels to replace with true labels in the target domain. If the quality of pseudo labels is low, can we still obtain good adaptation results in the target domain using the proposed method? If not, how does the quality of pseudo labels affects the performance of DA methods?
3.	In Theorem 1, this paper claims that Comp() decreases with larger observation numbers. However, it is still unknown if the Rademacher complexity (regarding deep networks) will converge to zero when we have infinite observations. This means that the claim made in the main content is not true, which will mislead broader ICLR readers. This claim is true under a fixed hypothesis with finite VC dimension (as said at the top of Page 20), which is not linked with your deep-network-based algorithm.
4.	The motivation of this paper is unclear. Why should we use theory based on Wasserstein distance? How about directly using pseudo labels (like SHOT in ICML20)? There are many questions regarding the motivation of this paper.
5.	The proposed method can only be used to address three transfer scenarios (presented in this paper)? Are there difficulties to apply the proposed method to address other DA problems? The motivation (testing the proposed method in the presented three problem settings) is unclear.
6.	The format of this paper is poor, which should be revised before submission.

---

> ### Author Response · Authors · 2020-11-18
> **Response 1/3**
>
> Thank you for your assessment and helpful comments, we hope the following responses can address your concerns. If you have additional questions, we are happy to provide more explanations!
>
> #### 1. Significance of main theoretical results: Theorem 1 and 2.
> > The main theorem (Theorem 1) assumes that we can get the label information in the target domain, which is not realistic in many DA problem settings.
> * In the multi-source UDA theories such as Ref. [a.1-a.4], they all had an *unobservable term* $\eta$ in the bound. Thus an underlying assumption for these theories is that a small $\eta$ is necessary to guarantee a successful UDA. However, Ref. [e.4] pointed out that $\eta$ can be very large when label shift occurs, indicating the limitation of current multi-source UDA theories for handling label shift.
> * However $\eta$ is impossible to estimate in UDA without labels [a.2; b.6-b.9; e.8], therefore Ref. [a.2] pointed out that in order to theoretically take $\eta$ into consideration in label shifts with different conditional distributions, a small amount of target labelled data is necessary.
> * Moreover, understanding the theory of multi-source transfer learning with limited target labels itself is an important research area that has recently appeared with many theoretical results Ref. [b.1-b.9] and practical attempts Ref. [c.1-c.8]. In particular, **Ref. [b.6-b.9] argued the practical/theoretical importance and challenge of measuring the divergence between two domains given label information for the target domain.** Theorem 1 extends this research line by adopting computationally feasible Wasserstein distance to explicitly handle the label shift (Section 3 in the main paper). Theorem 2 proposed new theoretical insights in the context of representation learning on latent space $z$, which remains elusive for the previous multi-source transfer theories such as Ref. [a.1-a.4], Ref. [b.6-b.9].
>
> * Besides, the *inspired* principles from Theorem 1 and 2 can be practically extended to the multi-source unsupervised DA, and multi-source partial UDA. We have discussed these techniques and required assumptions in Sec 4.1-4.2 and Question about pseudo-labels.
>
> #### 2. Question about pseudo labels
> > If the quality of pseudo labels is low, can we still obtain good adaptation results in the target domain using the proposed method?
>
> We additionally visualized the evolution of the pseudo-label prediction of different tasks shown in Appendix N (Figure 11) in the paper. From Figure 11, it can be observed that our algorithm can progressively improve the quality of pseudo-labels during training. In other words, our algorithm can *self-correct* inaccurate pseudo-labels and therefore is insensitive to the low quality of pseudo-labels at the early stage of the training process.
>
> > Theoretically analyse the effect of pseudo labels
>
> From the theoretical aspect of the proposed approach for UDA, we use the pseudo-label for estimating the label distribution ratio $\hat{\alpha}_t$ (Sec 4.2). Based on recent theoretical work Ref. [e,1-e,2] in label shift, this can have theoretical guarantee under additional assumptions with $$ || \hat{\alpha}_t - \alpha_t ||_2 \to 0 $$
>
> > How about directly using pseudo labels (like SHOT in ICML20)?
>
> SHOT is a technique of producing pseudo-labels, *whereas our theoretical framework and algorithm is about new generic principles in multi-source transfer learning for handling label shift.* In this work, we used the output of the network [e,1-e,2] to produce pseudo-labels. In other words, the techniques of pseudo-labels such as SHOT are in fact orthogonal to our framework and can be seamlessly integrated into our algorithm.
>
> #### 3. Rademacher Complexity in the deep neural-network
> > It is still unknown if the Rademacher complexity (regarding deep networks) will converge to zero when we have infinite observations. This means that the claim made in the main content is not true, which will mislead broader ICLR readers.
>
> * Thanks for your comments. In fact, the Rademacher complexity can be non-vacuously bounded in the deep neural-network by using recent theoretical results such as Ref. [d.1-d.5]. We can adopt these techniques analogously in our paper to obtain a meaningful bound in neural-network.
>
> * We also would like to emphasize that analyzing Rademacher complexity in deep neural networks is not our main focus in this work. Moreover, a large number of related works such as Ref. [a.1-a.4; b.6; b.9; e.1-e.7] relied on Rademacher complexity/Covering number/Metric Entropy for the theoretical analysis in DA but applied the inspired principles in deep neural networks for transfer learning.

---

> > ### Author Response · Authors · 2020-11-18
> > **Response 2/3**
> >
> > #####  4. Motivations
> >
> > > Why should we use theory based on Wasserstein distance?
> >
> > As Ref.[e.6] mentioned that W-distance is **theoretically tighter** than TV distance [e.4] and the commonly used Jensen-Shannon divergence [a.1]. According to theoretical analysis in [e,6], we have:
> >
> > $$W_1(P,Q) \leq C D_{TV}(P,Q) \leq C \sqrt{2D_{JS}(P,Q)}$$
> >
> > Where C is a positive constant. Therefore using Wasserstein distance can recover the previous theoretical results. In addition, Wasserstein distance takes into account the geometrical information of the distribution, which can be advantageous over other divergences [a.3, e.6].
> >
> > ###### High-level idea
> > > There are many questions regarding the motivation of this paper.
> >
> > We have revised our paper accordingly and the following summarizes our motivations.
> >
> > Labels shift problem commonly exists in many multi-source transfer learning problems. Recently [e.3; a.2; e.4] pointed out that conventionnel transfer algorithm by merely matching $D(S_t(z)\|T(z))$ and minimizing source empirical error is unable to handle label shift.
> >
> >  Therefore in our paper:
> >
> > * We proposed different principles with matching conditional distribution and $\alpha_t$ weighted-loss;
> >
> > * We proposed a guideline to aggregate different sources, based on the task relatedness, which remains elusive in previous work;
> >
> > * We proposed unified principles for three scenarios, while the previous multi-source approaches only handle one scenario [e.3-e.7].
> >
> > ##### 5. Motivation of considering three scenarios.
> >
> > > The proposed method can only be used to address three transfer scenarios (presented in this paper)? Are there difficulties to apply the proposed method to address other DA problems?
> >
> > Thanks for your question. We would like to emphasize that **most multi-source transfer works focused on only one scenario** such as multi-source transfer with limited target labels or multi-source unsupervised DA.
> >
> > In our paper, we proposed a *unified* framework that accommodates **three commonly encountered** multi-source transfer learning scenarios (transfer with limited target labels, UDA, and partial UDA).  i.e. Testing on these three scenarios is because they are widely studied in the previous DA works.  In fact, our principles *can be trivially* extended to other scenarios such as multi-source semi-supervised DA.
> >
> > In addition, our proposed principles demonstrate strong empirical results in all three scenarios with a good explainability when label shift occurs. E.g. Our proposed principle can exactly detect the non-overlapped classes in partial DA, which verifies the benefits of our principles.

---

> > > ### Author Response · Authors · 2020-11-18
> > > **Response 3/3**
> > >
> > > References:
> > >
> > > [a] Theories on multi-source UDA
> > >
> > > [a.1] Adversarial multiple source domain adaptation. Zhao, et.al NeurIPS 2018
> > >
> > > [a.2] Domain aggregation networks for multi- source domain adaptation. Wen, et.al ICML 2020
> > >
> > > [a.3] Extracting relationships by multi-domain matching. Li, et.al NeurIPS 2018
> > >
> > > [a.4] Moment matching for multi-source domain adaptation. Peng, ICCV 2019
> > >
> > > [b] Theories on transfer/multi-source transfer with limited target labels
> > >
> > > [b.1] Learning from Multiple Sources. Crammer.et.al JMLR 2008
> > >
> > > [b.2] Generalization Bounds for Domain Adaptation. Zhang.et.al NeurIPS 2012
> > >
> > > [b.3] New analysis and algorithm for learning with drifting distributions. Mohri.et.al ALT 2012
> > >
> > > [b.4] Flexible Transfer Learning under Support and Model Shift Wang.et.al NeurIPS 2014
> > >
> > > [b,5] Hypothesis transfer learning via transformation functions. Du.et.al NeurIPS 2017
> > >
> > > [b.6] Robust Learning from Untrusted Sources. Konstantinov.et.al ICML 2019
> > >
> > > [b.7] On the Value of Target Data in Transfer Learning. Hanneke.et.al NeurIPS 2019
> > >
> > > [b.8] Transfer Learning via Minimizing the Performance Gap Between Domains Wang.et.al NeurIPS 2019
> > >
> > > [b.9] A theory of multiple-source adaptation with limited target labeled data. Mansour,et.al 2020
> > >
> > > [c] Practice on transfer/multi-source transfer with limited target labels
> > >
> > > [c.1] Boosting for transfer learning with multiple sources. Yao.et.al CVPR 2010
> > >
> > > [c.2] Multisource transfer learning with convolutional neural networks for lung pattern analysis. IEEE journal of biomedical and health informatics, 21(1):76–84, 2016.
> > >
> > > [c.3] Source-target similarity modelings for multi-source transfer gaussian process regression. Wei.et.al ICML 2017
> > >
> > > [c.4] Transfusion: Understanding transfer learning for medical imaging. Raghu.et.al NeurIPS 2019
> > >
> > > [c.5] Transfer learning in natural language processing Ruder.et.al NAACL 2019
> > >
> > > [c.6] Multi-source cross-lingual model transfer: Learning what to share Chen.et.al ACL 2019
> > >
> > > [c.7] Semi-supervised Domain Adaptation via Minimax Entropy Saito.et.al ICCV 2019
> > >
> > > [c.8] Learning new tricks from old dogs: Multisource transfer learning from pre-trained networks Lee.et.al NeurIPS 2019
> > >
> > >
> > >
> > > [d] Rademacher complexity on Neural-Network
> > >
> > >
> > > [d.1] Data-dependent Sample Complexity of Deep Neural Networks via Lipschitz Augmentation. NeurIPS 2019
> > >
> > > [d.2] Algorithm-Dependent Generalization Bounds for Overparameterized Deep Residual Networks. NeurIPS 2019
> > >
> > > [d.3] Size-Independent Sample Complexity of Neural Networks. COLT 2018
> > >
> > > [d.4] Spectrally-normalized margin bounds for neural networks. NeurIPS 2017
> > >
> > > [d.5]Learning and generalization in overparameterized neural networks, going beyond two layers. NeurIPS 2019
> > >
> > > [e] Other thoretical results
> > >
> > > [e.1] A Unified View of Label Shift Estimation Garg, et.al 2020
> > >
> > > [e.2] Regularized learning for domain adaptation under label shifts. Azizzadenesheli, et.al ICLR 2019
> > >
> > > [e.3] On target shift in adversarial domain adaptation. Li, et.al Aistats 2019
> > >
> > > [e.4] Domain adaptation with conditional distribution matching and generalized label shift Combes, et.al. 2020
> > >
> > > [e.5] Bridging Theory and Algorithm for Domain Adaptation Zhang,et.al ICML 2019
> > >
> > > [e.6] Theoretical Analysis of Domain Adaptation with Optimal Transport. Redeo, et.al ECML 2017
> > >
> > > [e.7] Domain-Adversarial Training of Neural Networks. Ganin, et.al JMLR 2016
> > >
> > > [e.8] Impossibility Theorems for Domain Adaptation Ben David, et.al Aistats 2010

---

> > > > ### Comment · AnonReviewer4 · 2020-11-23
> > > > **Thanks for your reply.**
> > > >
> > > > This paper claims that unified principles for multi-source transfer learning under label shifts are proposed. However, without considering the quality of pseudo labels, the theoretical bound proposed in this paper cannot really provide unified principles. For example, how to theoretically analyze the relation between the quality of pseudo labels and the accuracy of estimation of W distance used in this paper? If we cannot answer this question, how to ensure that W distance is the best choice? Why not adversarial training or MMD distance? We cannot get real unified principles without considering the quality of pseudo labels, right?
> > > >
> > > > Making ambitious claims should be done carefully and examined by theoretical and practical results in detail. Otherwise, future researchers in this field will be misled.

---

> > > > > ### Author Response · Authors · 2020-11-24
> > > > > **Thanks for your comments**
> > > > >
> > > > > Thanks for your comments! We hope that the following explanations can address your concerns. If you have additional questions, we are happy to provide explanations.
> > > > >
> > > > > 1.  General response
> > > > >
> > > > > We think there might exist some miscommunications about unified principles. In our paper, the unified principles motivated by Theorem 1 and Theorem 2 are referred as:
> > > > >
> > > > > -- properly estimating label distribution ratio $\hat{\alpha}_t$
> > > > >
> > > > > -- learning task relations $\lambda$
> > > > >
> > > > > -- learning representation and classifier $g, h$
> > > > >
> > > > > In fact, our theorems, and principles rely on the target label information, so there is no such issue regarding the quality of the pseudo-labels. Please note that we do not claim that these inspired principles are about explicit algorithms for producing pseudo-label. We have revised our paper accordingly to clarify this point.
> > > > >
> > > > > In the unsupervised scenarios, your concern is correct in the sense that the algorithm can fail *in the worst-case*. We totally agree with you on this point. On the other hand, please note that our principles do not contradict your point since the **pseudo labels algorithms are complementary to our principles**. The key point is that the target labels cannot be estimated properly, the upper bound in our theorems can be quite loose. In this case, the theorems themselves and our principles of learning $\hat{\alpha}_t, \lambda, g$ and $h$ still hold, though the target risk can be poor due to the loose upper bound, which is consistent with your point.
> > > > >
> > > > > According to the impossibility theory in UDA [1,2], *there does not exist a general/unified target risk upper bound when label shift and conditional shift simultaneously occurs.* I.e. *in the worst-case*, all the pseudo-label algorithms will fail without target labels and additional assumptions. For example, one can use SHOT to generate pseudo labels to realize our principles by explicitly assuming (forces) that data of different classes are well separated and classes are balanced, and implicitly assumes that conditional shift is small. Our methods follow the assumptions in matched semantic conditions with $P_s(z|y)=P_t(z|y)$, and simply use the output of the network [3,4] to generate pseudo labels (Note this is orthogonal to our principles, but other techniques (e.g., SHOT) are also applicable.)
> > > > >
> > > > > Besides, the theoretical analysis is feasible when adopting proper algorithms and *additional assumptions*. In our paper, this can have theoretical analysis. (see Response to specific questions for details)
> > > > >
> > > > > It is also worth mentioning that we evaluate the proposed approaches through extensive experiments, and our methods consistently outperform other baselines in all scenarios, which verifies the utility and correctness of our framework. Our additional experiments in Appendix N (Figure 11) illustrated that our algorithm can progressively improve the quality of pseudo-labels during training. In other words, our algorithm can `self-correct`  inaccurate pseudo-labels and therefore is insensitive to the low quality of pseudo-labels at the early stage of the training process.
> > > > >
> > > > >
> > > > > 2. Response to specific questions
> > > > >
> > > > >
> > > > >
> > > > > > how to theoretically analyze the relation between the quality of pseudo labels and the accuracy of estimation of W distance used in this paper?
> > > > >
> > > > >   Thanks for your questions.
> > > > >
> > > > > 1.  In our proposed approach, through Lemma 1, the conditional W distance is equal to label-ratio $\alpha_t$ re-weighted adversarial loss.
> > > > >
> > > > > 2.  Therefore the quality in estimating the conditional W-distance is related to the quality of label distribution ratio $\hat{\alpha}_t$. i.e. If $\hat{\alpha}_t$ is equal to the ground truth $\alpha_t$, we can correctly estimate the conditional W distance through adversarial training.
> > > > >
> > > > > 3.  The $\hat{\alpha}_t$ is estimated through *the pseudo labels* and source information. Thus we need to guarantee the estimated ratio will converge to the ground truth $\hat{\alpha}_t\to \alpha_t$ in unsupervised DA.
> > > > >
> > > > > 4.  In Sec 4.2, the derived approach for estimating $\hat{\alpha}_t$ can have such a theoretical result with $\hat{\alpha}_t\to \alpha_t$ under additional assumptions $P_s(z|y)=P_t(z|y)$ see ref [3,4] for the theoretical conclusion.
> > > > >
> > > > > > Making ambitious claims should be done carefully and examined by theoretical and practical results in detail. Otherwise, future researchers in this field will be misled.
> > > > >
> > > > >  Thanks for your remark. We have revised our paper to clarify this. We would like to point out that our claims are examined by extensive practical results and ablation studies. As for theoretical results, we would like to emphasize that we clarified its exact theoretical settings, assumptions, goals, and how it motivates practice. (Sec 3, 4 in the paper)

---

### Decision · Program_Chairs · 2021-01-07
**Final Decision**

**Decision:**

Reject

**Comment:**


The question the authors address is relevant and interesting mostly in the UDA setting. However, there exists several recent works that have
highlighted the importance of label distribution ratio in DA (Wu et al., Combes et al. etc.), hence the main contribution of the
paper is to propose a novel analysis and results in the multi-source setting. That said, the paper has mixed reviews and
after going through the paper, the reviews and the discussion, I tend to agree with some of the reviewers that while
the idea is interesting, the paper lacks in several points that makes it unsuitable to publication, for now.

Here are the main points leading to the decision.


A) UDA is usually the most frequent situation that occurs in domain adaptation and the most difficult to handle.
The theoretical novelty of the bound comes only from the multi-source aspect that seems to be original

B) there is a strong contradiction in the paper. In the intro, they state that the paper addresses situations where conditional distributions differ. However in 4.2 they assume that they are finally equal.

In section 4.1, the authors show that for optimizing their problem, they need to have labels, mostly for estimating the class-conditional distributions. When these labels are available in the target domain, the problem is pretty simple and there exists many simple baselines that can handle this problem. However, in a UDA setting, they do not have label and proposes a method for estimation label proportion by assuming S_t(z|y) = T(z|y), which is in contradiction with their initial hypotheses S_t(z|y) != T(z|y). Hence under their assumption, the left hand side of Lemma 1 is zero and the equality is useless.
Hence, I would suggest the authors to avoid such a contradiction.

Under equality of S_t(z|y) = T(z|y), the approach proposed by the authors bears strong similarity with the work of Redko et al 2019 (cited in their paper). So I would highly to recommend them to compare with that algorithm. .


C) the authors use a lot a trick related to filtering, moving average.... I guess those parts is important for making the approach works and they are not properly analyzed.

D) The paper is  confusing in its writing and somehow this confusion makes the theoretical details hard to understand.
For instance, in section 3 the loss function is defined as having two variables but used one line after with only 1. In the theorem, it is not clear whether the true labelling function intervenes or how the y in h(x,y) is related to the true
labels. I guess a clarification is needed here for making the soundness of the theoretical results.